# A convenient Keldysh contour for thermodynamically consistent perturbative and semiclassical expansions

Vasco Cavina[1], Sadeq S. Kadijani[1], Massimiliano Esposito[1] and Thomas L. Schmidt[1,2]

**1** Department of Physics and Materials Science, University of Luxembourg,
L-1511 Luxembourg, Luxembourg
**2** School of Chemical and Physical Sciences, Victoria University of Wellington,
P.O. Box 600, Wellington 6140, New Zealand

★ vasco.cavina@uni.lu

## Abstract

The work fluctuation theorem (FT) is a symmetry connecting the moment generating functions (MGFs) of the work extracted in a given process and in its time-reversed counterpart. We show that, equivalently, the FT for work in isolated quantum systems can be expressed as an invariance property of a modified Keldysh contour. Modified contours can be used as starting points of perturbative and path integral approaches to quantum thermodynamics, as recently pointed out in the literature. After reviewing the derivation of the contour-based perturbation theory, we use the symmetry of the modified contour to show that the theory satisfies the FT at every order. Furthermore, we extend textbook diagrammatic techniques to the computation of work MGFs, showing that the contributions of the different Feynman diagrams can be added to obtain a general expression of the work statistics in terms of a sum of independent rescaled Poisson processes. In this context, the FT takes the form of a detailed balance condition linking every Feynman diagram with its time-reversed variant. In the second part, we study path integral approaches to the calculation of the MGF, and discuss how the arbitrariness in the choice of the contour impacts the final form of the path integral action. In particular, we show how using a symmetrized contour makes it possible to easily generalize the Keldysh rotation in the context of work statistics, a procedure paving the way to a semiclassical expansion of the work MGF. Furthermore, we use our results to discuss a generalization of the detailed balance conditions at the level of the quantum trajectories.

 Check for updates

# 1 Introduction

Green's functions (GFs) are commonly used to tackle problems of quantum transport in areas like molecular transport, electronics, superconductivity, nanojunctions and nuclear physics [1–5]. They can also be used to quantify the linear response to external perturbations, a fundamental tool to analyse experiments, for instance in magnetic resonance, Raman spectroscopy and crystallography [6–8]. Path integral techniques on the other hand, are convenient to study the relation between the quantum and the classical (stochastic) dynamics in open quantum systems [9–13]. They are also very suitable for studying the physics of phase transitions, instantons and critical phenomena in general [14,15]. In this seemingly heterogeneous landscape, the theoretical framework introduced by Keldysh [16–18] represents a unified way to

extend both, the usual Feynman approach to path integration and the equilibrium theory of GFs, to nonequilibrium systems [19,20] and has recently found applications in quantum and stochastic thermodynamics [21–24]. The idea behind the standard Schwinger-Keldysh technique is to simplify calculations by introducing an extended version of the time domain, called the Schwinger-Keldysh contour, that acts as a new ordered domain for the time variables of the theory. Different extensions of this formalism have proved to be suitable for computing charge and energy statistics using non-equilibrium GFs [23,25–33].

In the present paper, inspired by the extension of the contour idea to quantum thermodynamics in the papers of Funo and Quan [22] and Fei and Quan [24], we focus on the issue of thermodynamic consistency of perturbative and path integral approaches based on modified contours. Our first step is to introduce a modified Keldysh contour that can be used for calculating the work MGF in the evolution of an isolated driven quantum system. We then make use of the symmetries of the theory, in particular the FT [34–39], proving that it can be seen as a geometrical symmetry of the modified contour. This is crucial to the derivation of our other results. First, using the modified contour as a starting point for a perturbative expansion of the MGF, we show that its symmetry can be used to prove that the expansion is thermodynamically consistent, that is, it satisfies the FT at every perturbative order.

Then, following textbook diagrammatic techniques, we introduce an approach to the perturbative expansion of the MGF based on Feynman diagrams. The structure of the diagrammatic theory is similar to the one of standard perturbation theory [20,40], but while the architecture is the same, and we can picture it in terms of the topology of the diagrams, the building blocks, i.e., the propagators and the contour that acts as a domain of integration for the vertex variables, are different. As an application of this technique, we compute the work statistics when a small non-linear perturbation of a quadratic Hamiltonian is switched on between two energy measurements performed at two different times, showing that the probability distribution can be expressed as a linear combination of Poisson processes. We then analyze the thermodynamic consistency at the level of single Feynman diagrams and prove that the FT can also be interpreted as a detailed balance condition relating each diagram to its time-reversed counterpart.

We proceed in the second part of the paper by generalizing the Feynman path integral technique [41] to the modified contour. We use a similar strategy as in Ref. [22], but our approach is more suitable for defining a generalization of the Keldysh rotation (used to obtain a semiclassical expansion of the path integral formulation of dissipative systems [19,42–45]) in the context of quantum thermodynamics. We do so by introducing a *symmetrization* of the modified contour. After performing the symmetrization, we obtain a semiclassical expansion of the MGF where we explicitly compute the zeroth (classical) order and the first quantum correction to work fluctuations. Another benefit of the symmetrized contour approach is that it enables us to express the fluctuating work as a function of the endpoints of the quantum trajectories. This can be used to discuss the concept of detailed balance at the quantum trajectory level.

This paper is organized as follows. In Sec. 2 we introduce the Keldysh contour and its extension to the modified contour to express the MGF. In Sec. 3 we investigate the work statistics and show how to obtain the FT as a symmetry of the modified contour. In Sec. 4 we introduce the perturbation theory in terms of Green's functions and use the perturbation expansion to obtain the MGF of work in weakly perturbed quantum systems. Finally, in Sec. 5 we apply path integral techniques on the modified contour to compute the MGF. By symmetrizing the contour, we derive the MGF of work, compare it with similar approaches in the literature, and discuss the semiclassical limit in the path integral expression of the fluctuating work.

## 2 The contour in a general two-point measurement

The Schwinger-Keldysh contour makes it possible to simplify and express in an elegant way the theory of nonequilibrium many-body quantum systems. The same idea can be applied to the context of counting statistics in a double measurement scheme. In this section we briefly discuss the idea behind the contour and its generalization to counting statistics, before focusing our attention on more specific problems, like the work MGF.

Many physical properties of open quantum systems are encoded in correlation functions between pairs of operators [46]. Here we will use them as a starting point to understand the intuition behind the Keldysh contour (taking a similar route as more complete textbook derivations, e.g., Ref. [20]). Given two observables $A_1, A_2$ with time arguments in the Heisenberg picture given by $t_f, t'$ with $t_f \geq t'$ we can introduce the correlation function

$$C_{A_1, A_2}(t_f, t') = \text{Tr}\big[\rho_0 A_1^{(H)}(t_f) A_2^{(H)}(t')\big]. \tag{1}$$

The quantity above can be written in the Schrödinger picture after writing out explicitly the evolution operators contained in $A_1^{(H)}$ and $A_2^{(H)}$

$$C_{A_1, A_2}(t_f, t') = \text{Tr}\big[\rho_0 U^\dagger(t_f, 0) A_1 U(t_f, t') A_2 U(t', 0)\big], \tag{2}$$

where

$$U(t', t) = U^\dagger(t, t') = \mathcal{T} \exp\left(-\frac{i}{\hbar} \int_t^{t'} H(s) ds\right) = \bar{\mathcal{T}} \exp\left(\frac{i}{\hbar} \int_t^{t'} H(s) ds\right), \tag{3}$$

with $\mathcal{T}$ and $\tilde{\mathcal{T}}$ representing the time-ordering and anti-time-ordering operator, respectively. If we expand the evolution operators in Eq. (2) we have to deal with three ordered products and the calculations can become cumbersome. It is however possible to simplify the expansion by introducing some "bookkeeping" time arguments $A_1 \to A_1(t_f)$, $A_2 \to A_2(t')$, and by letting the time-ordering operator act on $A_1(t_f), A_2(t')$. In this way, the last term of Eq. (2) reduces to

$$A_1 U(t_f, t') A_2 U(t', 0) = \mathcal{T}\{e^{-\frac{i}{\hbar} \int_0^{t_f} dt H(t)} A_1(t_f) A_2(t')\}. \tag{4}$$

It is natural to ask if we can further simplify Eq. (2) by including also $U^\dagger(t_f, 0)$ in the time-ordered product in Eq. (4). A possible issue arises from the fact that the time arguments in $U^\dagger(t_f, 0)$ are the same as $U(t_f, t')$ and $U(t', 0)$, however, the position of these operators in the r.h.s. of Eq. (2) are evidently different. An elegant way to circumvent this problem is to introduce a new ordering operator $\mathcal{T}_K$, which orders the operators according to the position of their time arguments on a new domain $\gamma_K$, called the *Schwinger-Keldysh contour* [19,20,40,47]. This contour comprises two branches:

1. The forward branch $\gamma_-$, that goes from the initial time 0 to a final time $t_f$. The ordering operator $\mathcal{T}_K$ behaves like the usual time-ordering on this branch, so that, for instance, we can rewrite Eq. (4) as

$$A_1 U(t_f, t') A_2 U(t', 0) = \mathcal{T}_K\{e^{-\frac{i}{\hbar} \int_{\gamma_-} dt H(t)} A_1(t_f) A_2(t')\}. \tag{5}$$

2. The backward branch $\gamma_+$, that comes after the forward branch and goes back from the final time to 0 (see Fig. 1A). Since this branch of the contour goes backward in time, $\mathcal{T}_K$ is naturally sorting the operators from left to right with their time arguments increasing. This means that an anti-time-ordered product can be written as an ordered product over $\gamma_+$. For instance we have

$$U^\dagger(t_f, 0) = \mathcal{T}_K\{e^{-\frac{i}{\hbar} \int_{\gamma_+} dt H(t)}\}. \tag{6}$$

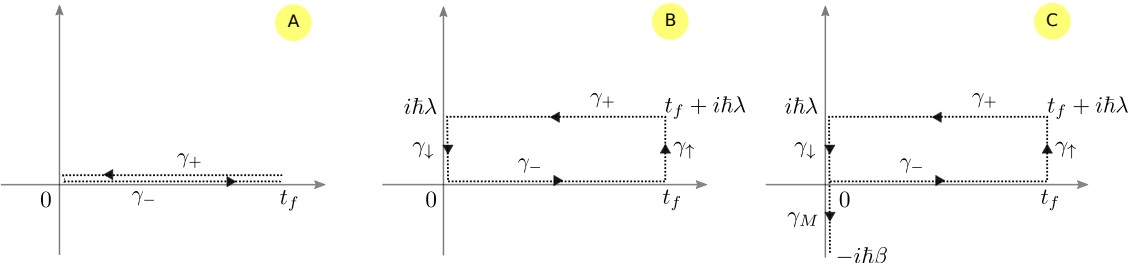

Figure 1: Three different time integration contours: A) The Keldysh contour $\gamma_K$, in which $\gamma_-$ and $\gamma_+$ denote, respectively, the forward and the backward branches. B) The contour $\gamma_C$, obtained by adding two vertical tracks $\gamma_{\uparrow,\downarrow}$ to the Keldysh contour. C) The contour $\gamma$, realized by augmenting the previous one with the track $\gamma_M$ encoding the initial condition.

If we put the arguments of $\mathcal{T}_K$ in Eqs. (5) and (6) under a single ordering, the operators with time argument on $\gamma_+$ will be placed to the left of those with time argument on $\gamma_-$ (since $\gamma_+$ follows $\gamma_-$ on the contour). In this way, $U^\dagger(t_f, 0)$ in Eq. (2) will be correctly placed to the left of $A_1$ and we finally obtain

$$C_{A_1, A_2}(t_f, t') = \text{Tr}[\rho_0 \mathcal{T}_K \{ e^{-\frac{i}{\hbar} \int_{\gamma_K} dt H(t)} A_1(t_f) A_2(t') \}]. \tag{7}$$

Note that the Hamiltonian $H(t)$ is now a function with argument in the contour $t \in \gamma_K$ that is equal in each branch to the physical Hamiltonian $H(t \in \gamma_-) = H(t \in \gamma_+)$.

Using an extension of the Keldysh idea, we can write the MGF in a double measurement process in a simple and compact way. In this framework a generic observable $\Lambda$ is measured at time 0 and time $t_f$ while the system evolves under the action of a time-dependent Hamiltonian between the two measurements. The generating function for the statistics of the difference of the outcomes of the first and second measurement is given by [37]

$$M_\Lambda(\lambda, t_f) = \text{Tr}[\rho_0 U^\dagger(t_f, 0) e^{\lambda \Lambda(t_f)} U(t_f, 0) e^{-\lambda \Lambda(0)}], \tag{8}$$

which is a special case of Eq. (1) with $A_1 = e^{\lambda \Lambda(t_f)}$ and $A_2 = e^{-\lambda \Lambda(0)}$, as well as $t' = 0$. Note that Eq. (8) assumes that the initial measurement operator $\Lambda(0)$ commutes with the initial state density matrix $\rho_0$, which is something we will always consider to be true. After expressing the MGF (8) in the form (7) and introducing the dummy integrations $e^{\lambda \Lambda(t_f)} = e^{-\frac{i}{\hbar} \int_0^{i\hbar\lambda} d\tau \Lambda(t_f)}$ and $e^{-\lambda \Lambda(0)} = e^{-\frac{i}{\hbar} \int_{i\hbar\lambda}^0 d\tau \Lambda(0)}$, we obtain

$$M_\Lambda(\lambda, t_f) = \text{Tr}\left\{ \rho_0 \mathcal{T}_K \left[ e^{-\frac{i}{\hbar} \int_{\gamma_K} dz H(z)} e^{-\frac{i}{\hbar} \int_0^{i\hbar\lambda} d\tau \Lambda(t_f)} e^{-\frac{i}{\hbar} \int_{i\hbar\lambda}^0 d\tau \Lambda(0)} \right] \right\}. \tag{9}$$

It is easy to see that the sum of the three integrals can be expressed as a single integration along a different contour $\gamma_C$ that consists in a modification of the Schwinger-Keldysh contour, in which two vertical branches are added at time 0 and $t_f$ (see Fig. 1B). Accordingly, the ordering operator also has to be extended to this new contour. This allows us to write

$$M_\Lambda(\lambda, t_f) = \text{Tr}\left\{ \rho_0 \mathcal{T}_C \left[ e^{-\frac{i}{\hbar} \int_{\gamma_C} dz H_C(z)} \right] \right\}, \tag{10}$$

where now the Hamiltonian $H_C$ is defined as

$$H_C(z) = \begin{cases} H(t) & \text{for } z = t \in \gamma_K, \\ \Lambda(t_f) & \text{for } z \in \gamma_\uparrow, \\ \Lambda(0) & \text{for } z \in \gamma_\downarrow, \end{cases} \tag{11}$$

where we labelled the vertical tracks of the contour drawn in Fig. 1B by $\gamma_{\uparrow,\downarrow}$. As a last step, we point out that the dependence on the initial state $\rho_0$ can also be cast as an integration over an additional track of the contour. Indeed, if we define an effective Hamiltonian $H_M$ by expressing $\rho_0$ as $\log \rho_0 = -\beta H_M - \log[Z(0)]$ with $Z(0) = \mathrm{Tr} e^{-\beta H_M}$, we can always write $\rho_0 = \exp\left[-\frac{i}{\hbar}\int_0^{-i\hbar\beta} dz H_M\right]/Z(0)$ and express Eq. (10) as in Ref. [24]

$$M_\Lambda(\lambda, t_f) = \frac{\mathrm{Tr}\left[\mathcal{T}_\gamma\left\{e^{-\frac{i}{\hbar}\int_\gamma dz H_\gamma(z)}\right\}\right]}{\mathrm{Tr}\left[\mathcal{T}_\gamma\left\{e^{-\frac{i}{\hbar}\int_\gamma dz H_{\gamma,0}(z)}\right\}\right]}, \tag{12}$$

where $\gamma$ is the contour defined in Fig. 1C, built by adding an additional track $\gamma_M$ along the imaginary axis, $\mathcal{T}_\gamma$ denotes the time-ordering operator along this contour, and the extended Hamiltonian $H_\gamma(z)$ is defined as

$$H_\gamma(z) = \begin{cases} H(t) & \text{for } z = t \in \gamma_K, \\ \Lambda(t_f) & \text{for } z \in \gamma_\uparrow, \\ \Lambda(0) & \text{for } z \in \gamma_\downarrow, \\ H_M & \text{for } z \in \gamma_M. \end{cases} \tag{13}$$

Moreover, we have defined $H_{\gamma,0} = H_\gamma|_{\Lambda=0}$. If the initial state is a Gibbs state, we have $H_M = H(0)$ and $\beta = 1/(k_B T)$ assumes the role of the inverse of the physical temperature $T$. Note that a similar modified contour can be derived for computing the characteristic function $M(i\lambda, t_f)$, but in this case the vertical branches of length $\lambda$ in Fig. 1C are replaced by horizontal ones [24]. We will discuss this point in more detail in the following sections (see Fig. 5).

## 3 Work fluctuation theorem as a symmetry of the contour

If the measurement operator $\Lambda(t)$ is the system Hamiltonian itself, the difference between the outcomes corresponds to the difference of the final and initial energies, i.e., the work performed on the system by an external source. The MGF of the work statistics is obtained from Eq. (8) by choosing $\Lambda(0) = H(0)$ and $\Lambda(t_f) = H(t_f)$,

$$M_W(\lambda, t_f) = \mathrm{Tr}\left[\rho_0 U^\dagger(t_f, 0)e^{\lambda H(t_f)}U(t_f, 0)e^{-\lambda H(0)}\right]. \tag{14}$$

Let us suppose now that the system is initialized in the Gibbs state, so that $H_M = H(0)$. In this setting it is possible to prove a fluctuation theorem (FT), a symmetry connecting the generating function $M_W$ and the generating function $M_W^{\mathrm{rev}}$ associated with the time-reversed work extraction process [35, 48]. We will now show that the FT can be seen as a symmetry of the contour $\gamma$.

We first present the standard derivation of the fluctuation theorem. To compute $M_W^{\mathrm{rev}}$, we first introduce the time-reversed process, in which the system is initialized in a Gibbs state corresponding to the final Hamiltonian, i.e., $\rho_{\mathrm{rev}}(0) = e^{-\beta H(t_f)}/Z(t_f)$, and then undergoes a time-reversed evolution. The latter is a combination of a time-reversal operation $\Xi$, which is an anti-unitary operator satisfying $\Xi i = -i\Xi$, and a forward evolution with the time-reversed driving protocol, i.e., $H_{\mathrm{rev}}(s) = H(t_f - s)$,

$$\Xi\rho_{\mathrm{rev}}(t_f)\Xi = U_{\mathrm{rev}}(t_f, 0)\Xi\rho_{\mathrm{rev}}(0)\Xi U_{\mathrm{rev}}^\dagger(t_f, 0), \tag{15}$$

where $U_{\mathrm{rev}}(t_f, 0) = \mathcal{T}\exp\left[-\frac{i}{\hbar}\int_0^{t_f} ds H_{\mathrm{rev}}(s)\right]$. Note that we suppose that the Hamiltonian does not depend on a magnetic field $B$, as such a case would lead to an additional minus sign,

$H_{\text{rev}}(s, B) = H(t_f - s, -B)$. Using Eq. (15) and $\Xi U^{\text{rev}}(t_f, 0)\Xi = U^\dagger(t_f, 0)$, we can deduce the following identity

$$
\begin{aligned}
M_W(\lambda, t_f) &= \frac{1}{Z(0)} \text{Tr}[U e^{-\beta H(0)} e^{-\lambda H(0)} U^\dagger e^{\lambda H(t_f)}] \\
&= \frac{1}{Z(0)} \text{Tr}[U e^{-\beta H(0)} e^{-\lambda H(0)} U^\dagger e^{(\lambda+\beta)H(t_f)} e^{-\beta H(t_f)}] \\
&= \frac{1}{Z(0)} \text{Tr}[U^\dagger e^{(\lambda+\beta)H(t_f)} e^{-\beta H(t_f)} U e^{-(\lambda+\beta)H(0)}] \\
&= \frac{Z(t_f)}{Z(0)} M_W^{\text{rev}}(-\beta - \lambda, t_f),
\end{aligned}
\tag{16}
$$

where we introduced the shorthand notation $U \equiv U(t_f, 0)$. After introducing the free energy difference as $Z(t_f)/Z(0) = e^{-\beta \Delta F}$, the above result leads to the fluctuation symmetry

$$
M_W(\lambda, t_f) = e^{-\beta \Delta F} M_W^{\text{rev}}(-\beta - \lambda, t_f).
\tag{17}
$$

To formulate this symmetry in the contour framework, we first note that the entire dependence of Eq. (12) on the counting field and the final time is established through the modified contour $\gamma$ in Fig. 1C. With this in mind, every step of the derivation (16) can be seen as a transformation of the contour itself as shown in Fig. 2. By making the $\lambda$ dependence explicit, this geometrical symmetry corresponds to $\gamma_\lambda = \gamma_{-\lambda-\beta}^{\text{rev}}$, where $\gamma^{\text{rev}}$ is the contour associated to the time-reversed generating function (panel D in Fig. 2). The weighted ratio between the generating functions of the forward and time-reversed processes becomes

$$
\frac{M_W(\lambda, t_f)Z(0)}{M_W^{\text{rev}}(-\lambda - \beta, t_f)Z(t_f)} = \frac{\text{Tr}[\mathcal{T}_\gamma \{ e^{-\frac{i}{\hbar} \int_{\gamma_\lambda} dz H_\gamma(z)} \}]}{\text{Tr}[\mathcal{T}_{\gamma_{-\lambda-\beta}^{\text{rev}}} \{ e^{-\frac{i}{\hbar} \int_{\gamma_{-\lambda-\beta}^{\text{rev}}} dz H_\gamma(z)} \}]} = 1,
\tag{18}
$$

which is equivalent to Eq. (17). The contour-based proof of the FT is a structural symmetry of any theory based on the contour $\gamma$ of Fig. 1C. Note that the idea of using a modification of the contour to encode symmetries that are intrinsic in the quantum theory has already proved to be a valid tool in non-equilibrium physics. For instance, outside the double energy measurement framework, it has been used to show that equilibrium theories share a fundamental symmetry at the level of the Schwinger-Keldysh action [49–52].

## 4 Thermodynamically consistent perturbation theory

The modified contour $\gamma$ can be used to build perturbative expansions of work generating functions in weakly perturbed quantum systems. Following the standard approach of perturbation theory, we decompose the contour Hamiltonian as $H_\gamma(z) = H_0(z) + \chi(z)H_1(z)$ where $H_0(z)$ and $H_1(z)$ are the unperturbed Hamiltonian and the perturbation, which are both defined on the contour $\gamma$, and $\chi(z)$ is a switching function. In complete analogy with Eq. (13), $H_1(z)$ has different physical interpretations depending on the position of $z$ on the contour $\gamma$: it represents an external perturbation of the physical Hamiltonian for $z = t \in \gamma_K$, while it is a correction to the measurement operator or to the initial state when we choose $z \in \gamma_{\uparrow,\downarrow}$ or $z \in \gamma_M$, respectively.

We will consider the work extraction in two scenarios. In the first scenario, the final Hamiltonian is equal to the initial one, i.e., $\chi(0) = \chi(t_f) = 0$. We will refer to this protocol as *switching on/off*, since the perturbation $H_1(z)$ is turned on after the first measurement and

switched off before the second one. This is the setting in which Bochkov and Kuzovlev derived the first integral fluctuation theorem (Eq. (8) in Ref. [53]) and the switching function in this case reads

$$\chi(z) = \begin{cases} \chi(t) & \text{for } z = t \in \gamma_K, \\ 0 & \text{for } z \in \gamma_{\uparrow,\downarrow}, \\ 0 & \text{for } z \in \gamma_M. \end{cases} \qquad (19)$$

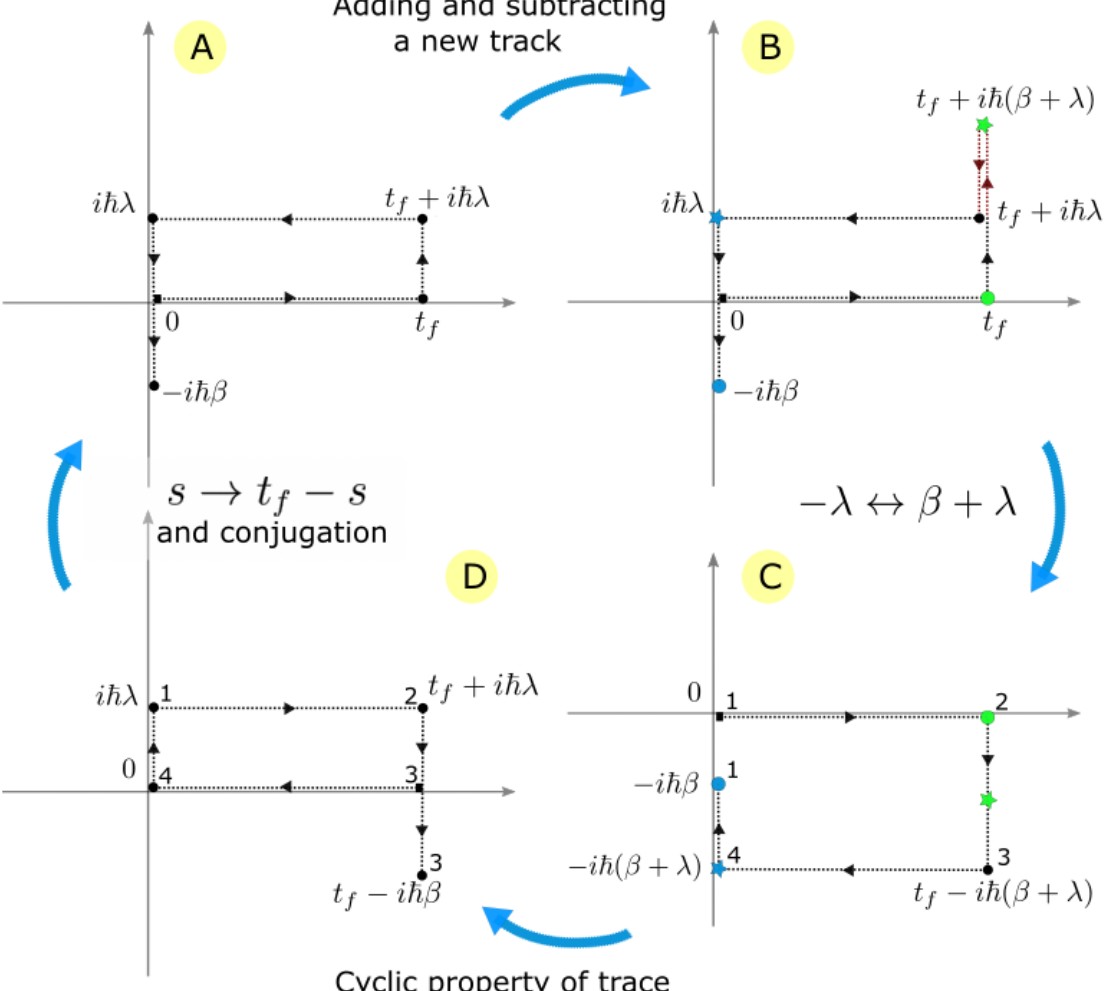

Figure 2: A graphical representation of the proof of Eq. (16). The correspondence between exponentials of time-dependent Hamiltonians and contour tracks allow us to establish a handy parallelism. Multiplying and dividing by $\exp[-\beta H(t)]$ is equivalent to adding and subtracting a new track ($A \to B$) in the picture. Exchanging $-\lambda$ with $\beta + \lambda$ can be expressed as an inversion of the blue/green stars with the blue/green circles ($B \to C$). The cyclic property of trace tells us that the starting point of the contour is arbitrary, so that if we mark the corners of the contour with $1, 2, 3, 4$, we can draw an equivalent contour starting from point 3 and running clockwise, instead of starting from point 1 ($C \to D$). Note that from point $C$ to $D$ we also shifted the contour upwards. The last step assumes time-reversal symmetry of the Hamiltonian and shows that the contour $D$ is indeed the one associated with the backward evolution of the original one ($D \to A$).

In the second scenario, the final Hamiltonian is arbitrary, i.e., $\chi(0) = 0$ but $\chi(t_f) \neq 0$. This is the general setting considered by Jarzynski [36] and the associated contour Hamiltonian is

$$\chi(z) = \begin{cases} \chi(t) & \text{for } z = t \in \gamma_K, \\ \chi(t_f) \neq 0 & \text{for } z \in \gamma_\uparrow, \\ 0 & \text{for } z \in \gamma_{M,\downarrow}. \end{cases} \tag{20}$$

The results about the perturbation theory and the diagrammatic expansion of the MGF discussed in the next subsections are true for both the protocols (19) and (20), while at the end of Sec. 4.2 we will continue our calculation by choosing specifically the protocol (19) and leave some comments about the generalization to the scenario (20) in App. D.

## 4.1 Interaction picture and non-interacting Green's functions

The customary approach to time-dependent perturbation theory begins with the introduction of the interaction picture [54]. To generalize this concept to the modified contour we use the following relation

$$U_\gamma(z, 0) \equiv \mathcal{T}_\gamma \{ e^{-\frac{i}{\hbar} \int_\gamma^z dz H_\gamma(z)} \} = U_{\gamma 0}(z, 0) \tilde{U}_{\gamma,1}(z, 0), \tag{21}$$

where we denoted with $\int_\gamma^z$ the integration between the initial point of the modified contour and the point $z$ and defined

$$U_{\gamma 0}(z, 0) = \mathcal{T} \{ e^{-\frac{i}{\hbar} \int_\gamma^z dz' H_0(z')} \}; \qquad \tilde{U}_{\gamma,1}(z, 0) = \mathcal{T} \{ e^{-\frac{i}{\hbar} \int_\gamma^z dz' \chi(z') \tilde{H}_1(z')} \}, \tag{22}$$

with the contour interaction Hamiltonian defined as $\tilde{H}_1(z) = U_{\gamma 0}(0, z) H_1(z) U_{\gamma 0}(z, 0)$ (see App. A for details). The l.h.s. of Eq. (21) evaluated in $-i\hbar\beta$ gives the numerator of (12), thus, after some manipulations we obtain

$$M_W(\lambda, t_f) = \frac{\text{Tr}[e^{-\beta H_M} \mathcal{T}_\gamma \{ e^{-\frac{i}{\hbar} \int_\gamma dz \chi(z) \tilde{H}_1(z)} \}]}{\text{Tr}[e^{-\beta H_M}]}, \tag{23}$$

where we used that $H_0(z) = H_M$ on the $\gamma_M$ branch of the contour. Equation (23) can also be written as an explicit function of $H_1(z)$

$$M_W(\lambda, t_f) = \frac{\text{Tr}[\mathcal{T}_\gamma \{ e^{-\frac{i}{\hbar} \int_\gamma dz H_0(z)} e^{-\frac{i}{\hbar} \int_\gamma dz \chi(z) H_1(z)} \}]}{\text{Tr}[\mathcal{T}_\gamma \{ e^{-\frac{i}{\hbar} \int_\gamma dz H_0(z)} \}]}. \tag{24}$$

The relevance of the equation above is due to the fact that when $H_1(z)$ is a small perturbation of $H_0(z)$, we can expand the second exponential in the numerator and obtain a perturbative series for the generating function. Before doing so let us focus first on the fundamental building block of perturbation theory which in systems of coupled bosons and fermions is given by the *Green's function*

$$G(z, z') \equiv -i \langle \mathcal{T}_\gamma \{ \tilde{c}(z) \tilde{c}^\dagger(z') \} \rangle = -i \text{Tr}[\mathcal{T}_\gamma \{ e^{-\frac{i}{\hbar} \int_\gamma dz H_0(z)} c(z) c^\dagger(z') \}], \tag{25}$$

where $c$ is a bosonic/fermionic annihilation operator and the angular brackets denote an average over the initial-state density matrix. Equation (25) is a straightforward generalization of the Keldysh contour GFs [19], but we stress that $z, z'$ are defined on the contour in Fig. 1C instead of the standard Keldysh contour. Using the evolution in the contour interaction picture (see App. A), the dynamical equation for the Green's function reads

$$\frac{d}{dz'} G(z, z') = \frac{1}{\hbar} \langle \mathcal{T}_\gamma \{ c(z) [H_0(z'), c^\dagger(z')] \} \rangle + i\delta(z - z'), \tag{26}$$

where the boundary conditions are given by the Kubo-Martin-Schwinger relations $G(z,0) = \pm G(z,-i\hbar\beta)$ for any $z \in \gamma$, where the upper or lower sign stands for bosons or fermions, respectively.

Let us now replace $H_0 = \hbar\omega c^\dagger c$. In this case Eq. (26) can be easily solved to obtain the non-interacting GFs (see Apps. B and B.1)

$$G_{b,f}^{(0)}(z,z') = -ie^{-i\omega(z-z')}\Big[ \pm n\Theta_\gamma(z'-z) + \bar{n}\Theta_\gamma(z-z')\Big], \tag{27}$$

where the upper (lower) sign again refers to bosons (fermions), $\bar{n} = 1 \pm n$, and $n = n_{b,f}(\omega) = (e^{\hbar\beta\omega} \mp 1)^{-1}$ denotes the Bose (Fermi) distribution function and $\omega$ is the associated frequency. In Eq. (27) we also introduced the contour step function $\Theta_\gamma(z-z')$, that is equal to 1 if $z$ is placed after $z'$ in the contour $\gamma$ and equal to 0 otherwise. As in standard Schwinger-Keldysh theory, the contour Green's function does not have an evident physical meaning, unlike its components which are obtained by restricting the domain of $z$ and $z'$ to selected branches of the contour (we use $\pm$ subscripts to denote time variables on the branches $\gamma_\pm$), for instance

$$G_{b,f}^{(0)}(t_+ + i\hbar\lambda, t'_-) = G_{b,f}^{(0)>}(t_+, t'_-) = -i\bar{n}e^{\hbar\omega\lambda}e^{-i\omega(t-t')},$$
$$G_{b,f}^{(0)}(t_-, t'_+ + i\hbar\lambda) = G_{b,f}^{(0)<}(t_-, t'_+) = \mp ine^{-\hbar\omega\lambda}e^{-i\omega(t-t')}, \tag{28}$$

are called greater ($>$) and lesser ($<$) components, the dependence by $\lambda$ arises from the vertical displacement in the complex plane between the $\gamma_-$ and $\gamma_+$ tracks. Other notable components can be obtained by choosing different contour arguments (see App. B.1), e.g., the time-ordered and anti-time-ordered components defined, respectively, as $G_{b,f}^{(0)T}(t_-, t'_-) \equiv G_{b,f}^{(0)}(t_-, t'_-)$ and $G_{b,f}^{(0)\bar{T}}(t_+, t'_+) \equiv G_{b,f}^{(0)}(t_+ + i\hbar\lambda, t'_+ + i\hbar\lambda)$, which have the property of being $\lambda$-independent.

## 4.2 Perturbative expansion and diagrams

Within the perturbative framework, we can expand the second integral in the numerator of Eq. (24) in terms of correlation functions of arbitrary order, and obtain an expression of the form (see also Ref. [24])

$$M_W(\lambda, t_f) = \sum_{n=0}^{\infty} \frac{1}{n!}\left(-\frac{i}{\hbar}\right)^n \int_\gamma dz_1 \dots dz_n \frac{\text{Tr}[\mathcal{T}_\gamma\{e^{-\frac{i}{\hbar}\int_\gamma dz H_0(z)}\chi(z_1)H_1(z_1)\dots\chi(z_n)H_1(z_n)\}]}{\text{Tr}[\mathcal{T}_\gamma\{e^{-\frac{i}{\hbar}\int_\gamma dz H_0(z)}\}]} . \tag{29}$$

At each order $n$ we have a $n$-point correlation function of the Hamiltonian $H_1$. If we assume that $H_1$ is a linear combination of products of fermionic and bosonic creation and annihilation operators, the correlation functions can be decomposed into non-interacting GFs following Wick's theorem [20, 40]. It is important to underline that the expansion of the correlation functions in terms of non-interacting GFs is the same as in standard perturbation theory because Wick's theorem is a consequence of the commuting (or anti-commuting) algebra of the operators $c, c^\dagger$ and as such holds independently of the contour of integration.

The discussion above can be reformulated in diagrammatic terms since the Feynman diagrams entering in the calculation of Eq. (29) are the same as in standard perturbation theory, the only difference being the contour of integration of the vertex variables $z_1, \dots, z_n$ and the non-interacting GFs themselves, which are given by Eq. (27). To illustrate this, let us consider as an example the second-order perturbation theory of the Hamiltonian $H_1 = \hbar\omega_\chi(a + a^\dagger)c^\dagger c$ where $c$ and $a$ represent, respectively, the annihilation operator of a fermionic and a bosonic mode. In this case the contribution of the second order ($n = 2$) to the numerator of Eq. (29)

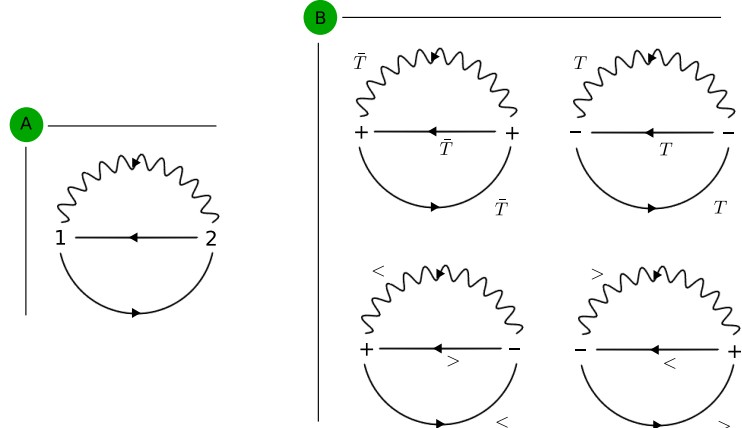

Figure 3: An example of how different Keldysh components appears in the diagram depicted on the left. The solid and wiggly lines denote the fermionic and bosonic non-interacting GFs, respectively.

will correspond to two connected Feynman diagrams, the one in Fig. 3A and the "dumbbell" diagram (see Fig. 6 in App. C). Focusing on the former, which we call $M_W^{(2),1}(\lambda, t)$, we have

$$M_W^{(2),1}(\lambda, t_f) = -i\omega_\chi^2 \iint_\gamma dz_1 dz_2 \chi(z_1)\chi(z_2) G_b^{(0)}(z_1, z_2) G_f^{(0)}(z_1, z_2) G_f^{(0)}(z_2, z_1). \tag{30}$$

Assuming a switching on/off protocol as in Eq. (19) and dividing the integration over $\gamma$ in an integration over $\gamma_-$ and $\gamma_+$ (the general rule to divide an integration over the contour in many integrals over the branches has been introduced by Langreth [55]), we rewrite the above integral in terms of the components of the GFs as

$$M_W^{(2),1}(\lambda, t_f) = -i\omega_\chi^2 \iint_0^{t_f} dt_1 dt_2 \chi(t_1)\chi(t_2)\times \tag{31}$$

$$\times \left\{ G_b^{(0)\bar{T}}(t_1, t_2) G_f^{(0)\bar{T}}(t_1, t_2) G_f^{(0)\bar{T}}(t_2, t_1) + G_b^{(0)T}(t_1, t_2) G_f^{(0)T}(t_1, t_2) G_f^{(0)T}(t_2, t_1) \right.$$

$$\left. - G_b^{(0)>}(t_1, t_2) G_f^{(0)>}(t_1, t_2) G_f^{(0)<}(t_2, t_1) - G_b^{(0)<}(t_1, t_2) G_f^{(0)<}(t_1, t_2) G_f^{(0)>}(t_2, t_1) \right\}.$$

In complete analogy to the connection between Eq. (30) and Fig. 3A, the four contributions to Eq. (31) can be expressed in terms of Feynman diagrams with a "charge" $\pm$ to take into account the position of the vertex variables on the contour, see Fig. 3B.

The time-ordered and the anti-time-ordered components are independent of $\lambda$ while $G^{<,>}$ include a factor of $e^{\pm\hbar\omega\lambda}$ which carries the frequency $\hbar\omega$ [see Eq. (28)]. This simple dependence allows us to introduce the energy transferred in a "charged" diagram $d$ as $E_d = \hbar \sum_i s_i^d \omega_i$ where $i$ runs over all the propagators and $s_i^d = 0, -1, 1$ depending on whether the propagator $i$ in the diagram $d$ is a $G^<$ function ($s_i^d = -1$), a $G^>$ function ($s_i^d = 1$), or a $G^{T,\bar{T}}$ function ($s_i^d = 0$). Each diagram will contribute to the generating function a factor $\Gamma_d(t_f)e^{\lambda E_d}$, where $\Gamma_d(t_f)$ is a prefactor that can be found by computing the contribution of the diagram (in the example of Eq. (31) by computing the associated double integral over the time variables).

It is convenient to introduce the cumulant generating function, given by the logarithm of Eq. (12),

$$C_W(\lambda, t_f) = \log \text{Tr}[\mathcal{T}_\gamma\{e^{-\frac{i}{\hbar}\int_\gamma dz H_\gamma(z)}\}] - \log \text{Tr}[\mathcal{T}_\gamma\{e^{-\frac{i}{\hbar}\int_\gamma dz H_0(z)}\}]. \tag{32}$$

The diagrammatic expansion of each of the two logarithms contains only connected diagrams, as a consequence of the linked cluster theorem (see for instance Ref. [20], chapter 11). The second logarithm contains exactly the same diagrams of the first one, but has $\lambda = 0$. As a result we obtain the following general formula for the cumulant generating function

$$C_W(\lambda, t_f) = \sum_{d,\text{conn.}} \Gamma_d(t_f)(e^{\lambda E_d} - 1), \tag{33}$$

where the summation variable $d$ runs over all the connected "charged" diagrams. We stress again that the result (33) is true when assuming that the switching function is non-zero only in the horizontal branches, that is Eq. (19). The cumulant generating function (33) can be interpreted as a sum of independent rescaled Poisson processes with average $\Gamma_d(t_f)$ and jumps in energy given by $E_d$. Therefore, every diagram represents a channel through which a quantized amount of energy $E_d$ can be exchanged with a Poissonian rate given by $\Gamma_d(t_f)$. The universality of Eq. (33) can be seen as a consequence of the Poisson law of rare events, since within the perturbative approach the rate of each energy exchange process is small. We expect Eq. (33) to break down when an infinite number of diagrams is resummed, in analogy to what happens in the case of charge statistics [26, 56].

In App. C we explore several applications of the result (33), by computing the rates $\Gamma_d(t_f)$ explicitly for some specific models. In the case of the Holstein coupling $H_1 = \hbar\omega_\chi(a + a^\dagger)c^\dagger c$ our results can also be verified by direct calculation of the cumulant generating function using the matrix form of the time evolution operator (see App. E).

We are now in a position to come back to our initial claim that the fluctuation theorem is satisfied at every order of the perturbative expansion. In Sec. 3 we showed that the contours used to compute $M_W(\lambda, t_f)$ and $M_W^{\text{rev}}(-\lambda - \beta, t_f)$ are the same. Since the propagators (see Eq. (27)) and the integration domain of the vertex variables (see Eq. (29)) are fully determined by the contour, the two perturbative expansions with the same contour are identical. This ensures, as announced, that for the switching-on/off scenario (19) the symmetry $M_W(\lambda, t) = M_W^{\text{rev}}(-\lambda - \beta, t)$ is preserved at every perturbative order.

## 4.3 Time-reversed diagrams and detailed balance

We can investigate the effects of the FT at the level of the single Feynman diagrams. For this sake, we have to connect the diagrams appearing in the perturbative expansion of the moment generating function $M_W(\lambda, t_f)$, with the ones appearing in the expansion of its time-reversed counterpart $M_W^{\text{rev}}(\lambda, t_f)$. The two MGFs share the same Hamiltonian, but the shape of the contour is different: while the vertex coordinates of the diagrams associated to $M_W(\lambda, t_f)$ live on $\gamma$ in Fig. 2 A, in the case of $M_W^{\text{rev}}$ they are defined on $\gamma^{\text{rev}}$ in Fig. 2 D. It is thus clear that the definition of the time-reversed diagrams goes through a mapping between the coordinates of the contours $\gamma$ and $\gamma^{\text{rev}}$. To understand the nature of this mapping, let us consider a "charged" diagram $d$ appearing in the decomposition of the MGF in the scenario (20) (see e.g. one of the four contributions to Eq. (31)). We introduce its time-reversed as the diagram $\bar{d}$ appearing in the expansion of $M_W^{\text{rev}}(\lambda, t_f)$ in which the sign of all the "charges" is the same as in $d$. Note that the positions of $\gamma_+$ and $\gamma_-$ are inverted in $\gamma^{\text{rev}}$ if compared to $\gamma$ (see also [31]), thus for any greater GF $G_{b,f}^{(0)>}(t, t')$ appearing in the formula for $d$, there will be a lesser GF $G_{b,f}^{(0)<}(t, t')$ appearing in $\bar{d}$ and vice versa.

Since from Eq. (28) we have

$$\frac{G_{b,f}^{(0)<}(t, t')}{G_{b,f}^{(0)>}(t, t')} = \pm e^{-\beta\hbar\omega} e^{-2\lambda\hbar\omega}, \tag{34}$$

we can conclude that the ratio between the contribution of a "charged" diagram and its time-reversed contains a factor $\pm e^{\beta\hbar\omega}e^{-2\lambda\hbar\omega}$ for every lesser GF and a factor $\pm e^{-\beta\hbar\omega}e^{+2\lambda\hbar\omega}$ for every greater GF contained in the original diagram, where $\omega$ is the frequency of the fermionic/bosonic mode associated with the propagator. After introducing $E_d$ as in Sec. 4.2 and setting $\lambda = 0$, we find that the weights $\Gamma_d(t_f)$ and $\Gamma_d^{\text{rev}}(t_f)$ of a given diagram and its time-reversed are related by

$$\frac{\Gamma_d(t_f)}{\Gamma_d^{\text{rev}}(t_f)} = e^{\beta E_d}, \tag{35}$$

where the sign contribution $\pm$ in Eq. (34) can be neglected by assuming that the interaction Hamiltonian contains an even number of fermionic fields. Equation (35) can be seen as a diagrammatic version of the detailed balance conditions [57–60]. Note that the condition (35) is stronger than the FT for the cumulant generating function, $C_W(\lambda, t_f) = C_W^{\text{rev}}(-\lambda - \beta, t_f)$, since it holds at a more detailed level, i.e., the level of single transitions. The FT at the level of the cumulant generating function can now be easily recovered by using Eqs. (35) and (33). To do so, we write the time-reversed generating function explicitly

$$C_W^{\text{rev}}(-\lambda - \beta, t_f) = \sum_{d,\text{conn.}} \Gamma_d^{\text{rev}}(t_f) e^{-(\lambda+\beta)E_d^{\text{rev}}} - \sum_{d,\text{conn.}} \Gamma_d^{\text{rev}}(t_f). \tag{36}$$

where $E_d^{\text{rev}}$ is the energy transferred in the time reversed charged diagrams, that satisfies $E_d^{\text{rev}} = -E_d$. Using the detailed balance relation in (35), we replace $\Gamma_d^{\text{rev}}(t_f) = \Gamma_d(t_f)e^{-\beta E_d}$ in the first term of Eq. (36) which gives

$$C_W^{\text{rev}}(-\lambda - \beta, t_f) = \sum_{d,\text{conn.}} \Gamma_d(t_f) e^{\lambda E_d} - \sum_{d,\text{conn.}} \Gamma_d^{\text{rev}}(t_f). \tag{37}$$

To write the second term as a function of the forward rates $\Gamma_d(t_f)$, we note that since the second addend on the right hand side of Eq. (32) is equal to both $\log Z(0)$ and the $\lambda$-independent term in Eq. (33), we have $\sum_{d,conn} \Gamma_d(t_f) = \log Z(0)$. In the time-reversed generating function, $Z(0)$ should be replaced by $Z(t_f)$ (see Sec. 3), however, the two quantities are equal due to the assumption (19) (switching on/off scenario). Therefore, we conclude that $\sum_{d,conn} \Gamma_d^{\text{rev}}(t_f) = \sum_{d,conn} \Gamma_d(t_f)$. Replacing this into Eq. (37) we conclude that $C_W^{\text{rev}}(-\lambda - \beta, t_f) = C_W(\lambda, t_f)$.

## 5 Path integration using the modified contour

Another useful approach to studying the generating function is to express Eq. (12) in terms of path integrals [22, 23, 61]. This approach is an extension of the usual Feynman path integral approach on the Keldysh contour [41, 62, 63]. We will see that our modified contour is particularly suitable for work statistics and for describing the semiclassical limit of the MGF by considering an expansion for small $\hbar$ of the generating function [43]. Let us consider the Hamiltonian of a single particle in an external potential,

$$H(t) = \frac{P^2}{2m} + V[\alpha(t), X], \tag{38}$$

where $P$ is the momentum operator, $m$ is mass and $V[\alpha(t), X]$ is a single particle potential in the particle position $X$, which depends parametrically on an external driving parameter $\alpha(t)$. Plugging the Hamiltonian (38) into Eq. (12) and performing a Trotter decomposition of the contour-ordered exponentials, we obtain the path integral form of the moment generating function,

$$M_W(\lambda, t_f) = \frac{1}{Z(0)} \int \mathcal{D}x(z)\mathcal{D}p(z) e^{\frac{i}{\hbar}S[x(z),p(z)]}, \tag{39}$$

where $p(z)$ and $x(z)$ are the momentum and position fields defined on the modified contour $\gamma$, $Z(0)$ is the partition function corresponding to the Hamiltonian at time 0 and $S$ is the classical action

$$S = \int_\gamma dz \left\{ \frac{dx(z)}{dz} p(z) - H_\gamma[x(z), p(z)] \right\}. \tag{40}$$

Here we present the calculation for a measurement operator corresponding to the total energy, $\Lambda = \frac{P^2}{2m} + V(X)$, but the formalism works similarly for other choices of the measurement operator. We split the fields on the contour by defining their components, in a similar way to what was done for the GFs. Defining $z = t + i\tau$ we have

$$x(z) = \begin{cases} x_-(t) & \text{for } z = t \in \gamma_-, \\ x_+(t) & \text{for } z = t + i\lambda \in \gamma_+, \\ x_\uparrow(\tau) & \text{for } z = t_f + i\tau \in \gamma_\uparrow, \\ x_\downarrow(\tau), x_M(\tau) & \text{for } z = i\tau \in \gamma_{\downarrow,M}, \end{cases} \tag{41}$$

and analogously for $p(z)$. Eliminating the momentum operator $p$ by integrating over $\mathcal{D}p(z)$, one obtains the Lagrangian representation of the path integral (see App. F)

$$M_W(\lambda, t_f) = \frac{1}{Z(0)} \int \mathcal{D}'x(z) e^{\frac{i}{\hbar} S[x(z)]}, \tag{42}$$

where $\mathcal{D}'x(z)$ is the new measure of integration following the elimination of the momenta, and the Lagrangian action reads

$$S = \int_\gamma dz \, \mathcal{L}_\gamma[\alpha(z), x(z)], \tag{43}$$

where $\mathcal{L}_\gamma$ is the Lagrangian on the modified contour that according to the portion of the contour of interest can assume different forms as

$$\mathcal{L}_\gamma[\alpha(z), x(z)] = \begin{cases} \mathcal{L}[\alpha(t), x_-(t)] & z = t \in \gamma_-, \\ \mathcal{L}[\alpha(t), x_+(t)] & z = t + i\lambda \in \gamma_+, \\ \mathcal{L}_\uparrow[\alpha(t_f), x_\uparrow(\tau)] & z = t_f + i\tau \in \gamma_\uparrow \\ \mathcal{L}_{\downarrow,M}[\alpha(0), x_{\downarrow,M}(\tau)], & z = i\tau \in \gamma_{\downarrow,M}, \end{cases} \tag{44}$$

where we have

$$\mathcal{L}[\alpha(t), x(t)] = \frac{1}{2} m \left[ \frac{dx(t)}{dt} \right]^2 - V[\alpha(t), x(t)],$$

$$\mathcal{L}_\uparrow[\alpha(t_f), x_\uparrow(\tau)] = -\frac{1}{2} m \left[ \frac{dx_\uparrow(\tau)}{d\tau} \right]^2 - V[\alpha(t_f), x_\uparrow(\tau)], \tag{45}$$

$$\mathcal{L}_{\downarrow,M}[\alpha(0), x_{\downarrow,M}(\tau)] = -\frac{1}{2} m \left[ \frac{dx_{\downarrow,M}(\tau)}{d\tau} \right]^2 - V[\alpha(0), x_{\downarrow,M}(\tau)].$$

This indicates that the Lagrangian on the vertical branches is the negative of the classical energy. The generating function in terms of the action in Eq. (40) can be used to study the characterization of the work fluctuations at the path integral level and its semiclassical limit.

## 5.1 Symmetrization of the contour and Keldysh rotation

Following the quantum-classical correspondence principle [64], the generating function in the semiclassical limit should reproduce its classical analog at first non-zero order in $\hbar$. This means that the path integral form of the generating function (42), in which fields are defined on the contour $\gamma$, should reproduce its stochastic path integral counterpart in a suitable limit [65–67]. The first obstacle that we have to overcome when trying to find such a correspondence is the difference between the domains of integration of the stochastic path integral, that is $[0, t_f]$ and the Keldysh contour [6, 68, 69]. In the path integral representation of the dynamics, this obstacle is simply overcome by the fact that the forward ($\gamma_-$) and the backward ($\gamma_+$) branches are equal, so the integration on the Keldysh contour can be seen as an integration over the segment $[0, t_f]$ of the difference between the forward and backward actions [19]. Applying this reasoning to the contour in Fig. 1C, we look for a transformation of the contour that makes it possible to divide it into two equal halves. This symmetrization is carried out by assigning half of the vertical lines of $\gamma_\uparrow$, $\gamma_\downarrow$ and $\gamma_M$ to an upper branch named $\gamma_\oplus$ and the other half of the lines to a lower branch $\gamma_\ominus$ (see Fig. 4 for a detailed explanation of this procedure). In the symmetrized contour in Fig. 4C, an argument $z \in \gamma_\oplus$ can be mapped to $\gamma_\ominus$ simply by complex conjugation. This allows us to write the action in Eq. (43) as

$$
\begin{aligned}
S &= \int_{\gamma_\ominus} \mathcal{L}_\gamma[x_\ominus(z), \alpha(z)]\,dz + \int_{\gamma_\oplus} \mathcal{L}_\gamma[x_\oplus(z), \alpha(z)]\,dz \\
&= \int_{\gamma_\ominus} \mathcal{L}_\gamma[x_\ominus(z), \alpha(z)]\,dz - \int_{\gamma_\ominus} \mathcal{L}_\gamma[x_\oplus(z^*), \alpha(z^*)]dz^* \\
&= \int_{\gamma_\ominus} \left\{ \mathcal{L}_\gamma[x_\ominus(z), \alpha(z)] - \mathcal{L}_\gamma[x_\oplus(z^*), \alpha(z^*)] \right\} d\,\mathrm{Re}\,z \\
&\quad + i \int_{\gamma_\ominus} \left\{ \mathcal{L}_\gamma[x_\ominus(z), \alpha(z)] + \mathcal{L}_\gamma[x_\oplus(z^*), \alpha(z^*)] \right\} d\,\mathrm{Im}\,z,
\end{aligned}
\tag{46}
$$

where in the last equality we separated the contribution of the real and imaginary parts of the differential $dz$. Interestingly, the contour in Fig. 4 has a great similarity with the the symmetric contour used by Aron et al. [49] to study the symmetries of the Schwinger-Keldysh action in equilibrium and non-equilibrium systems. We can now perform an analog of the Keldysh rotation [42–44, 70] and introduce the classical and quantum fields as

$$
x_{cl}(z) = \frac{1}{2}\left[ x_\ominus(z) + x_\oplus(z^*) \right], \qquad x_q(z) = \frac{1}{2}\left[ x_\ominus(z) - x_\oplus(z^*) \right].
\tag{47}
$$

Inserting the above expressions into the action in Eq. (46), we can rewrite the generating function as a path integral over $\gamma_\ominus$. Since the integrand of the action in Eq. (46) depends on the branch ($\mathrm{Im}\,z$ nullifies on $\gamma_{(\ominus,-)}$ while $\mathrm{Re}\,z$ nullifies on $\gamma_{(\ominus,\uparrow)}$, $\gamma_{(\ominus,\downarrow)}$ and $\gamma_{(\ominus,M)}$) it is convenient to separate the contributions of the horizontal and vertical branches, obtaining

$$
M_W(\lambda, t_f) = \frac{1}{Z(0)} \int \mathcal{D}'x_{cl/q}(z) e^{\frac{1}{\hbar}\int_{\gamma_{(\ominus,\downarrow),(\ominus,M)}} \Sigma[x_{cl}(z), x_q(z)]d\,\mathrm{Im}\,z}
$$
$$
\times\, e^{\frac{i}{\hbar}\int_{\gamma_{(\ominus,-)}} \mathcal{M}[x_{cl}(z), x_q(z)]d\,\mathrm{Re}\,z}\, e^{\frac{1}{\hbar}\int_{\gamma_{(\ominus,\uparrow)}} \Sigma[x_{cl}(z), x_q(z)]d\,\mathrm{Im}\,z}.
\tag{48}
$$

where $\mathcal{D}'x_{cl/q}(z) = \mathcal{D}'x_{cl}(z)\mathcal{D}'x_q(z)$ and we introduced the functions

$$
\begin{aligned}
\Sigma &= -m\left( \dot{x}_{cl}^2 + \dot{x}_q^2 \right) + V(\alpha, x_{cl} + x_q) + V(\alpha, x_{cl} - x_q), \\
\mathcal{M} &= 2m\dot{x}_{cl}\dot{x}_q - V(\alpha, x_{cl} + x_q) + V(\alpha, x_{cl} - x_q).
\end{aligned}
\tag{49}
$$

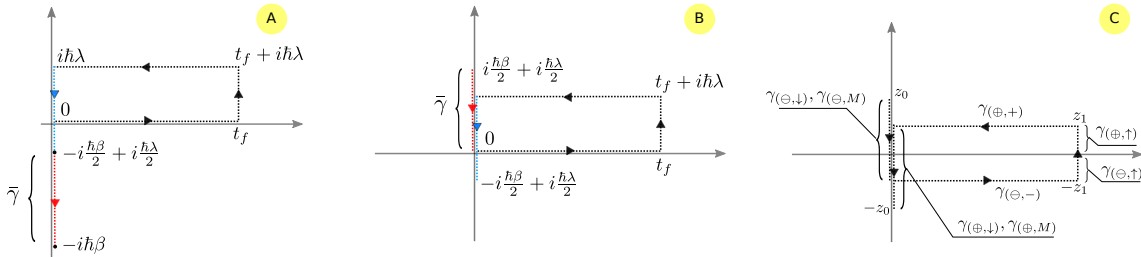

Figure 4: The equivalence between the contour of Fig. 1C and a new symmetric contour. A) The last interval $[-i\frac{\hbar\beta}{2} + i\frac{\hbar\lambda}{2}, -i\hbar\beta]$, denoted with $\bar{\gamma}$ in the picture, can be removed and attached to the initial part of the contour using the cyclic property of trace. The result of this operation is represented in the next panel. B) We can translate along the imaginary axis by $\frac{\hbar\lambda}{2}$ using the invariance of the generating function under such a translation, thus obtaining the contour in the next panel. C) The final symmetric version of the contour. In this picture, we have used $z_0 = i\frac{\hbar\beta}{2}$ and $z_1 = t_f + i\frac{\hbar\lambda}{2}$. The $\gamma_{\uparrow,\downarrow,M}$ branches of the contour in Fig. 1C can be naturally divided into two parts in this representation, which we denote by $\gamma_{(\ominus,\uparrow)}$, $\gamma_{(\oplus,\uparrow)}$, by $\gamma_{(\ominus,\downarrow)}$, $\gamma_{(\oplus,\downarrow)}$ and by $\gamma_{(\ominus,M)}$, $\gamma_{(\oplus,M)}$ respectively. The branch $\gamma_{(\ominus,M)}$ goes from $z_0$ to $z = 0$, while the branch $\gamma_{(\ominus,\downarrow)}$ goes from $z = 0$ to $z = -i\frac{\hbar\lambda}{2}$.

Notice that in Eq. (46) the domain of the fields of the path integral is $\gamma_\ominus$, with starting and ending points given by $z = i\beta\hbar/2$ and $z = t_f$. Since in these points the forward and backward fields are equal, the boundary conditions for Eq. (48) are $x_q(t_f) = x_q(i\beta\hbar/2) = 0$. It is now natural, taking inspiration from the classical case in which the fluctuating work can be defined as a function of the endpoints of the stochastic trajectories, to introduce a quantum energy function, defined at time $t_f$, as

$$E_Q(\lambda, t_f) = \frac{1}{\hbar\lambda} \int_{\gamma_{(\ominus,\uparrow)}} \Sigma[x_{cl}(z), x_q(z)] d \operatorname{Im} z, \tag{50}$$

and its analogue at the initial time $t = 0$, where $\gamma_{(\ominus,\uparrow)}$ is replaced by $\gamma_{(\ominus,\downarrow)}$. The difference between the initial and final energy functions gives a characterization of the fluctuating work at the trajectory level, and we can write the MGF as

$$M_W(\lambda, t_f) = \frac{1}{Z(0)} \int \mathcal{D}' x_{cl/q}(z) e^{\frac{1}{\hbar} \int_{\gamma_{(\ominus,M)}} \Sigma(z) d \operatorname{Im} z} e^{\frac{i}{\hbar} \int_{\gamma_{(\ominus,-)}} \mathcal{M}(z) d \operatorname{Re} z} e^{\lambda W(\lambda, t_f)}, \tag{51}$$

where $\mathcal{M}(z)$ and $\Sigma(z)$ are given in Eq. (49) and

$$W(\lambda, t_f) = E_Q(\lambda, t_f) - E_Q(\lambda, 0). \tag{52}$$

The specific choice of the symmetrization shown in Fig. 4 is essential to obtain a representation of the MGF in which $W(\lambda, t_f)$ depends only on the initial and final points of the trajectories. If we had chosen a different contour instead of considering the contour in Fig. 4C and its lower half $\gamma_\ominus$ as the domain of integration in the last of Eqs. (46), $W(\lambda, t_f)$ would have acquired a different functional dependence on the fields. A particularly interesting choice is the one used by Funo and Quan [22] that we will summarize in the next section.

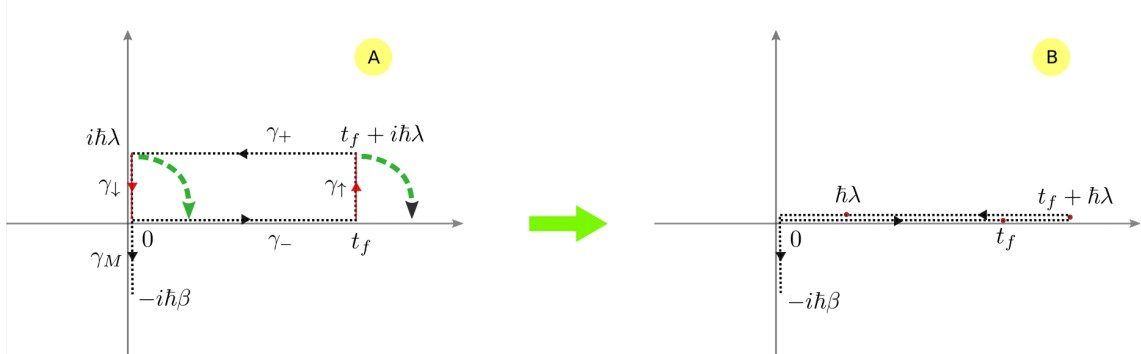

Figure 5: Transforming the modified contour to the asymmetric one by changing $\lambda \to -i\lambda$. A) the modified contour. B) Asymmetric contour resulting from the transformation. One can use this contour to define the work functional in terms of the forward paths [22].

## 5.2 Connection with the results of Funo and Quan

To compare our approach with the one presented in Ref. [22], we assume as a first step that the counting field is purely imaginary and replace $\lambda \to -i\lambda$. This amounts to replacing the contour $\gamma$ with the flat one in Fig. 5B. Such a contour allows a calculation of the characteristic function of work, which using path integrals can be written as

$$M_W(-i\lambda, t_f) = \frac{1}{Z(0)} \int \mathcal{D}' x_M \mathcal{D}' x \mathcal{D}' y \, e^{-\frac{i}{\hbar}(S_1[x] - S_2[y]) - \frac{1}{\hbar} S_M[x_M]}, \tag{53}$$

where $S_1$, $S_2$ and $S_M$ are the the actions of the forward, backward and $\gamma_M$ branches in the contour of Fig. 5B, respectively,

$$S_1 = \int_0^{\hbar\lambda} ds \mathcal{L}[\alpha(0), x(s)] + \int_{\hbar\lambda}^{\hbar\lambda + t_f} ds \mathcal{L}[\alpha(s - \hbar\lambda), x(s)],$$

$$S_2 = \int_0^{t_f} ds \mathcal{L}[\alpha(s), y(s)] + \int_{t_f}^{\hbar\lambda + t_f} ds \mathcal{L}[\alpha(t_f), y(s)],$$

$$S_M = \int_0^{-\hbar\beta} ds \mathcal{L}_M[\alpha(0), x_M(s)]. \tag{54}$$

The fields on the path integral (53) have boundary conditions $x(\hbar\lambda + t_f) = y(\hbar\lambda + t_f)$, $x_M(-\hbar\beta) = x(0)$, and $x_M(0) = y(0)$. These conditions are a direct consequence of the continuity of the field $x(z)$ in the modified contour, since $x, y, x_M$ are its components in the contour in Fig. 5B (analogously to the case of the contour in Fig. 5A in which they are given by Eq. (41)). Note that differently from the contour in Fig. 5B, the Lagrangian in the forward and backward branches are now different. This lays behind the definition of the work functional as a difference between the forward and backward action in terms of the forward path $x$ (for details we refer to Ref. [22])

$$i\lambda W_\lambda[x] = \frac{i}{\hbar}\left(S_1^\lambda[x] - S_2^\lambda[x]\right), \tag{55}$$

which after some manipulations makes it possible to express the work functional at the level of quantum trajectories:

$$W_\lambda = \int_0^{t_f} dt \frac{1}{\hbar\lambda} \int_0^{\hbar\lambda} ds\, \dot\alpha(t) \frac{\partial V[\alpha(t), x(t+s)]}{\partial\alpha(t)}, \tag{56}$$

where the integrand of the first integral acts as a quantum generalization of the instantaneous power. This result is different from the representation (52) in which the work is expressed as a function of the endpoints of the trajectories only.

## 5.3 Semiclassical limit

To analyze the semiclassical limit it is convenient to represent the path integral in the Hamiltonian convention, retaining the integral over the $p$ fields until the end of the calculation. In this case, we can repeat the procedure in Sec. 5.1 and obtain the same result as in Eq. (48) but with the path integral now including the momentum variables $\mathcal{D}p_{x/cl}$. The functions $\mathcal{M}$ and $\Sigma$ are replaced by (see App. G.1 for details)

$$\Sigma_h = \mathcal{K} + m\left(p_{cl}^2 + p_q^2\right) + V[\alpha, x_{cl} + x_q] + V[\alpha, x_{cl} - x_q],$$

$$\mathcal{M}_h = \mathcal{K} + \frac{2}{m}p_{cl}p_q - V[\alpha, x_{cl} + x_q] + V[\alpha, x_{cl} - x_q], \tag{57}$$

where $\mathcal{K} = 2p_q(dx_{cl}/dz) - 2x_q(dp_{cl}/dz)$. Since we expect the final energy to be a function of $x_{cl}(t_f), p_{cl}(t_f)$ in the classical limit, we will assume that the contribution of $x_q$ and $p_q$ is small on the vertical branches. This intuition can be verified by keeping the nonlinear contributions in $x_q$ and $p_q$, and showing that they scale as an higher power of $\hbar$ (see App. G and [43]).

To first order in $p_q$ and $x_q$, Eq. (57) leads to $\Sigma_h \approx \mathcal{K} + 2E_{cl}(z) = \mathcal{K} + p_{cl}^2(z)/m + 2V[\alpha(t_f), x_{cl}(z)]$. Therefore, at this order, the path integrals over $x_q$ and $p_q$ become trivial. Since the function $\mathcal{K}$ contains the first order of the quantum fields, the path integral on $\gamma_{(\ominus,\uparrow)}$ is written as

$$\int \mathcal{D}x_q \mathcal{D}p_q e^{\int_{\gamma_{(\ominus,\uparrow)}} \mathcal{K}(x_{cl}, x_q, p_{cl}, p_q) dz} = \prod_{\xi=p,q} \delta[\xi_{cl}(z) - \xi_{cl}(t_f)]. \tag{58}$$

Hence, the only admissible classical paths in this limit are the ones in which $x_{cl}(z), p_{cl}(z)$ in the vertical branches are always equal to $x_{cl}(t_f), p_{cl}(t_f)$. With this in mind, the path integral over the classical fields can be easily carried out,

$$\int \mathcal{D}x_{cl} \mathcal{D}p_{cl} e^{\int_{\gamma_{(\ominus,\uparrow)}} [\frac{1}{m}p_{cl}^2 + 2V(\alpha, x_{cl})] dz} \prod_\xi \delta[\xi_{cl}(z) - \xi_{cl}(t_f)] = e^{\lambda\left\{\frac{p_{cl}^2(t_f)}{2m} + V[\alpha(t_f), x_{cl}(t_f)]\right\}}. \tag{59}$$

Combining this result with the one we obtain doing the analogous calculation on $\gamma_{(\ominus,\downarrow)}$ and $\gamma_{(\ominus,M)}$, we finally conclude that

$$W(\lambda, t_f) = E_{cl}(t_f) - E_{cl}(0) + O(\hbar). \tag{60}$$

that is the classical version of Eq. (52). This result is independent of the counting field $\lambda$ and coincides with work fluctuations in classical driven isolated systems [36, 71]. We obtained Eq. (60) as a limit of Eq. (52), but we could have obtained it as the classical limit of Eq. (56) as well. In the latter case it can be shown (see [22]) that the term inside the time integral in Eq. (56) reduces to the instantaneous power output generated by the time-dependent driving and Eq. (60) is recovered after carrying out the time integration.

The first quantum correction to Eq. (60) can be obtained by keeping the quadratic terms in $x_q$ and $p_q$ in the expansion of $\Sigma_\hbar$. In this case we are left with a Gaussian integral over the quantum fields, which we computed explicitly in App. G.3. If we introduce $W(\lambda, t_f) = W_0(t_f) + \lambda W_1(t_f) + \lambda^2 W_2(t_f) + O(\lambda^3, \hbar^2)$ with $W_0(t_f) = E_{cl}(t_f) - E_{cl}(0)$, we have

$$
\begin{aligned}
W_1(t_f) &= -\frac{\hbar^2 V''[\alpha(t_f), x_{cl}(t_f)]}{4m} + \frac{\hbar^2 V''[\alpha(0), x_{cl}(0)]}{4m} \\
W_2(t_f) &= \frac{\hbar^2 V''[\alpha(t_f), x_{cl}(t_f)]}{12m} \left[ \frac{p_{cl}^2(t_f)}{2m} + \frac{V'^2[\alpha(t_f), x_{cl}(t_f)]}{2V''[\alpha(t_f), x_{cl}(t_f)]} \right] \\
&\quad - \frac{\hbar^2 V''[\alpha(0), x_{cl}(0)]}{12m} \left[ \frac{p_{cl}^2(0)}{2m} + \frac{V'^2[\alpha(0), x_{cl}(0)]}{2V''[\alpha(0), x_{cl}(0)]} \right],
\end{aligned}
\tag{61}
$$

where $W_1$ contributes to the variance of work, that corresponds to the second derivative of the generating function with respect to $\lambda$. Moreover, the above results become exact for time-dependent harmonic potentials (see App. H). In this case, we recover the results of Ref. [72].

## 5.4 Detailed balance and connections with the Wigner function

One of the main advantages of the representation of the work (51), (52) arising from the contour in Fig. 4C is that while the information on the dynamics is all contained in the forward branch $\gamma_{(\ominus,-)}$, the $\lambda$ dependence is isolated in the vertical branches. This separation between the dynamical contribution to the action and the "thermodynamical" one allows us to solve, at least formally, both contributions separately. This is not possible if we represent the work as in Eq. (56), since the fields appearing in the definition of $W_\lambda$ are the same appearing in the dynamical action (i.e., the fields in the forward branch). We can exploit this advantage to study a generalization of the concept of detailed balance at the level of quantum trajectories, in a way that is conceptually similar to what we did in Sec. 4.3 in the case of the diagrammatic approach to perturbation theory. As a first step, we introduce the formal solution to the path integral on the vertical branch $\gamma_{(\ominus,\uparrow)}$, that is the last part of Eq. (48)

$$
\mathcal{E}_{\lambda, t_f}[y_q^f, y_{cl}^f, \alpha(t_f)] = \frac{1}{\lambda} \log \int \mathcal{D}' x_{cl/q} e^{\frac{1}{\hbar} \int_{\gamma_{(\ominus,\uparrow)}} \Sigma[x_{cl}(z), x_q(z)] d\,\mathrm{Im}\,z},
\tag{62}
$$

where the boundary conditions of the path integral are $x_{q\uparrow}(0) = 0$, $x_{q\uparrow}(-\hbar\lambda/2) = y_q^f$, and $x_{cl\uparrow}(-\hbar\lambda/2) = y_{cl}^f$, and we made the dependence on $\alpha(t_f)$ explicit for convenience. Note that, in a similar way to Eq. (41), $x_{q\uparrow}(\tau)$ and $x_{cl\uparrow}(\tau)$ are defined as the quantum and classical fields on the branch $\gamma_{(\ominus,\uparrow)}$, by replacing $z = t + i\tau$. Since the contribution of the other vertical branch $\gamma_{(\ominus,\downarrow),(\ominus,M)}$ is functionally the same, apart from replacing $\lambda \to -\lambda - \beta$, we can write Eq. (48) as

$$
M_W(\lambda, t_f) = \frac{1}{Z(0)} \int dy_q^f dy_{cl}^f dy_q^i dy_{cl}^i \frac{e^{\lambda \mathcal{E}_\lambda[y_q^f, y_{cl}^f, \alpha(t_f)]}}{e^{(\beta+\lambda)\mathcal{E}_{-\lambda-\beta}[y_q^i, y_{cl}^i, \alpha(0)]}} \mathcal{U}[y_q^f, y_{cl}^f, t_f; y_q^i, y_{cl}^i, 0],
\tag{63}
$$

with $\mathcal{U}$ representing the propagator from the initial values of the fields $(y_q^i, y_{cl}^i)$ to the final values $(y_q^f, y_{cl}^f)$, that can be obtained by integrating the contribution of the forward branch in Eq. (48) with the appropriate boundary conditions (see also Eqs. (62), (143), (144) of App. I for details). Let us define the integrand of Eq. (48) for $\lambda = 0$ as

$$
K(y_q^f, y_{cl}^f, y_q^i, y_{cl}^i) = Z(0)^{-1} \mathcal{U}[y_q^f, y_{cl}^f, t_f; y_q^i, y_{cl}^i, 0] e^{-\beta \mathcal{E}_{-\beta}[y_q^i, y_{cl}^i, \alpha(0)]}.
\tag{64}
$$

It has a similar structure as the joint probability distribution of a classical system with a two-dimensional configuration space spanned by $(y_q, y_{cl})$, prepared in a Gibbs state with Hamiltonian $\mathcal{E}_{-\beta}[y_q^i, y_{cl}^i, \alpha(0)]$, and following a stochastic evolution that brings the system in $(y_q^f, y_{cl}^f)$. However, since $\mathcal{U}$ is in general a complex number, $K$ lacks a probabilistic interpretation. Despite this, we can give an interpretation to Eq. (64) in terms of Wigner functions [42,43,73–75]. Indeed it is possible to prove that

$$\int dy_q^i dy_q^f K(y_q^f, y_{cl}^f, y_q^i, y_{cl}^i) = C \int dp^i dp^f \Pi(p^f, y_{cl}^f, p^i, y_{cl}^i) W_\beta(p^i, y_{cl}^i), \tag{65}$$

where $W_\beta$ is the Wigner representation of the initial state, $\Pi$ is the Weyl transform of the propagator (see App. I) and $C$ is a normalization constant. It is known that $W_\beta$ and $\Pi$ on the r.h.s. of Eq. (65) reduce, respectively, to a classical Gibbs state distribution in the phase space, and to a classical propagator that imposes the deterministic equations of motion in the phase space [73]. We can use both the integrand of the l.h.s. and the integrand of the r.h.s. in Eq. (65) to discuss the concept of detailed balance. Let us introduce $K^{\mathrm{rev}}$ as the integrand of Eq. (48) for $\lambda = 0$, but in the symmetrized version of the time-reversed contour $\gamma^{\mathrm{rev}}$ in Fig. 2D. In the time-reversed contour with $\lambda = 0$ the only vertical branch is placed at $Rez = t_f$, and represents the initial preparation of the time-reversed trajectories. With this in mind, it is easy to prove that

$$\frac{K(y_q^f, y_{cl}^f, y_q^i, y_{cl}^i)}{K^{\mathrm{rev}}(-y_q^i, y_{cl}^i, -y_q^f, y_{cl}^f)} = \frac{e^{-\beta\mathcal{E}_{-\beta}[y_q^i, y_{cl}^i, \alpha(0)]}}{e^{-\beta\mathcal{E}_{-\beta}[-y_q^f, y_{cl}^f, \alpha(t_f)]}} e^{-\beta\Delta F}, \tag{66}$$

where $e^{-\beta\Delta F} = Z(t_f)/Z(0)$ and the minus sign in front of the quantum variables at the denominator comes from the fact that the forward and backward branches are inverted in the time-reversed contour. As discussed above, $K$ is not necessarily real and has no operational meaning in terms of measurements. On the contrary, if we choose $y_q^i = y_q^f = 0$ it represents the joint probability distribution $P(y_{cl}^f, y_{cl}^i)$ in a two-point measurement process of the *position operator*, with initial and final measured values given by $y_{cl}^i$ and $y_{cl}^f$ (see App. I). In the classical limit, this reduces to

$$\frac{P(y_{cl}^f, y_{cl}^i)}{P^{\mathrm{rev}}(y_{cl}^i, y_{cl}^f)} = e^{\beta[V(\alpha(t_f), y_{cl}^f) - V(\alpha(0), y_{cl}^i)] - \beta\Delta F} + O(\hbar), \tag{67}$$

that is, a detailed balance condition for the exchanges of the potential energy of the system. This is expected since in the initial measurement there is no information on the initial momentum of the particle. To obtain the classical form of the detailed balance in the phase space representation, we have to rely on the r.h.s. of Eq. (65) instead. Indeed, even if the Wigner function is non-positive in general, it is in the classical limit, where we obtain

$$\frac{\Pi(p^f, y_{cl}^f, p^i, y_{cl}^i) W_\beta(p^i, y_{cl}^i)}{\Pi^{\mathrm{rev}}(-p^i, y_{cl}^i, -p^f, y_{cl}^f) W_\beta^{\mathrm{rev}}(-p^f, y_{cl}^f)} = \frac{W_\beta(p^i, y_{cl}^i)}{W_\beta^{\mathrm{rev}}(-p^f, y_{cl}^f)} \approx e^{\beta(W_0 - \Delta F)} + O(\hbar), \tag{68}$$

where $W_0$ is given by Eq. (60), in analogy with the classical results [71], and $W_\beta^{\mathrm{rev}}$ denotes the Wigner function associated to the initial state of the time-reversed trajectory. The last equality also follows from the equivalence of the Wigner function and classical Gibbs state in the classical limit [73]. The minus signs in front of the final momentum has been added for the sake of completeness, in our calculations the Hamiltonian is always quadratic in $p$ and the change of sign becomes irrelevant.

# 6  Conclusions

We used the modified Keldysh contour as a versatile tool to investigate the perturbative approach to MGFs and their semiclassical limit. The symmetry property of the extended contour proved instrumental in showing the consistency of the perturbative expansion, ensuring that the fluctuation theorem is satisfied at every order in the expansion of the MGF. We showed that the modified contour technique is particularly suitable for diagrammatic approaches, and we used it to derive the work distribution in a switch on/off scenario, showing the work to be distributed as a linear combination of rescaled Poisson processes. Each of those processes represent a channel in which a discrete package of energy is exchanged between the system and the experimental driving apparatus. To investigate the semiclassical limit of the MGF, we expressed it in terms of a Feynman path integral and discussed how the form of the action depends on the choice of the modified contour. By symmetrizing the contour, we used the Keldysh rotation to write the action in terms of the classical and quantum fields, in a generalization of the Keldysh rotation approach. Our technique allows for a discussion of the detailed balance conditions, both at the level of the diagrammatic and the path integral approaches. In the former case, we show how the fluctuation theorem can be seen as resulting from a stronger detailed balance symmetry at the level of the single diagrams, while in the latter case we proposed a way to generalize the detailed balance to quantum trajectories and showed how this approach allows us to make contact with the Wigner function and with classical phase space approaches to thermodynamics. A interesting future perspective would be to generalize our approach to many-body open quantum systems [3,20] and assess the advantages of our new contour. This should be particularly relevant for preserving thermodynamic consistency while calculating work and heat counting statistics using known approximations schemes such as GW or random phase (RPA) [46,76,77]. To conclude, we expect that generalizing our formalism to the case in which the system is in contact with many baths at different temperatures will present a very interesting challenge, since in this extended scenario the temperature is not unique, nor is the counting field, considering that one is typically interested in measuring many different thermodynamic fluxes (e.g. the different heat flows in each one of the reservoirs). We expect that studying in detail the many baths scenario could help in bridging the non-equilibrium GF formalism and the modified contour formalism with other general approaches to thermodynamic consistency in stochastic thermodyanmics [59].

# Acknowledgements

**Funding information**    VC amd SSK acknowledge financial support by the National Research Fund Luxembourg under the grant CORE QUTHERM C18/MS/12704391.

# A  Interaction picture on the modified contour

The standard approach to time-dependent perturbation theory is based on the interaction picture [54]. We start by proving Eq. (21). The derivative of $U_\gamma(z,0)$ in respect to $z$ reads

$$\frac{d}{dz}U_\gamma(z,0) = -\frac{i}{\hbar}H_\gamma(z)U_\gamma(z,0).$$ (69)

The preceding equation can be shown by doing an infinitesimal displacement of the generator and expanding to first order

$$U_\gamma(z+\epsilon,0) \equiv \mathcal{T}_\gamma\{e^{-\frac{i}{\hbar}\int_\gamma^{z+\epsilon} H_\gamma(z)dz}\} = \left[\mathbb{I} - i\frac{\epsilon}{\hbar}H_\gamma(z)\right]\mathcal{T}_\gamma\{e^{-\frac{i}{\hbar}\int_\gamma^z H_\gamma(z)dz}\} + O(\epsilon^2)$$

$$\left[\mathbb{I} - i\frac{\epsilon}{\hbar}H_\gamma(z)\right]U_\gamma(z,0) + O(\epsilon^2).$$ (70)

Let us now compute the derivative of the right hand side of Eq. (21) and verify that is the same as Eq. (70).

$$\frac{d}{dz}U_{\gamma 0}(z,0)\tilde{U}_{\gamma,1}(z,0) = -iH_0(z)U_{\gamma 0}(z,0)\tilde{U}_{\gamma,1}(z,0) - i\chi(z)U_{\gamma 0}(z,0)\tilde{H}_1(z)\tilde{U}_{\gamma,1}(z,0)$$

$$= -iH_0(z)U_{\gamma 0}(z,0)\tilde{U}_{\gamma,1}(z,0) - i\chi(z)U_{\gamma 0}(z,0)\tilde{H}_1(z)U_{\gamma 0}(0,z)U_{\gamma 0}(z,0)\tilde{U}_{\gamma,1}(z,0)$$

$$= -iH_0(z)U_{\gamma 0}(z,0)\tilde{U}_{\gamma,1}(z,0) - i\chi(z)H_1(z)U_{\gamma 0}(z,0)\tilde{U}_{\gamma,1}(z,0),$$ (71)

and the last term is equal to the r.h.s. of (69) after regrouping $H_0(z)$ and $\chi(z)H_1(z)$. Note now that the numerator of Eq. (12) in terms of $U_\gamma(z,0)$ is simply given by

$$\text{Tr}\left[\mathcal{T}_\gamma\{e^{-\frac{i}{\hbar}\int_\gamma H_\gamma(z)dz}\}\right] = \text{Tr}[U_\gamma(-i\hbar\beta,0)],$$ (72)

and we can use Eq. (21) to write

$$\text{Tr}[U_\gamma(-i\hbar\beta,0)] = \text{Tr}[U_{\gamma 0}(-i\hbar\beta,0)\tilde{U}_{\gamma,1}(-i\hbar\beta,0)] = \text{Tr}[e^{-\beta H_0}\tilde{U}_{\gamma,1}(-i\hbar\beta,0)].$$ (73)

After expanding $\tilde{U}_{\gamma,1}$ as a contour ordered exponential and dividing by the proper normalization, we obtain Eq. (23) of the main text.

# B  Green's functions and higher order correlation functions

A generic two-point correlation function on the contour in the Schrodinger picture is written as

$$C_{A_1,A_2}(z_1,z_2) \equiv \langle\mathcal{T}_\gamma\{A_1^{(H)}(z_1)A_2^{(H)}(z_2)\}\rangle$$

$$= \Theta_\gamma(z_1-z_2)\langle A_1^{(H)}(z_1)A_2^{(H)}(z_2)\rangle + \Theta_\gamma(z_2-z_1)\langle A_1^{(H)}(z_2)A_2^{(H)}(z_1)\rangle$$

$$= \Theta_\gamma(z_1-z_2)\text{Tr}[U_\gamma(-i\hbar\beta,z_1)A_1(z_1)U_\gamma(z_1,z_2)A_2(z_2)U_\gamma(z_2,0)]$$

$$+ \Theta_\gamma(z_2-z_1)\text{Tr}[U_\gamma(-i\hbar\beta,z_2)A_2(z_2)U_\gamma(z_2,z_1)A_1(z_1)U_\gamma(z_1,0)].$$ (74)

To ease the calculations we rewrite the two-point correlation function as

$$C_{A_1,A_2}(z_1,z_2) = \Theta_\gamma(z_1-z_2)\text{Tr}[U_\gamma(-i\hbar\beta,0)U_\gamma(0,z_1)A_1(z_1)U_\gamma(z_1,z_2)A_2(z_2)U_\gamma(z_2,0)]$$

$$+ \Theta_\gamma(z_2-z_1)\text{Tr}[U_\gamma(-i\hbar\beta,0)U_\gamma(0,z_2)A_2(z_2)U_\gamma(z_2,z_1)A_1(z_1)U_\gamma(z_1,0)].$$ (75)

To take the the derivatives with respect to $z_1$ and $z_2$, we use the equations bellow which come from (70)

$$\frac{d}{dz_1}U_\gamma(z_1,z_2) = -\frac{i}{\hbar}H_\gamma(z_1)U_\gamma(z_1,z_2)$$

$$\frac{d}{dz_1}U_\gamma(z_2,z_1) = \frac{i}{\hbar}U_\gamma(z_2,z_1)H_\gamma(z_1). \tag{76}$$

Using above equations we can take the derivative of (75) to obtain the equations of motion for the two-point correlation function

$$\frac{\partial}{\partial z_1}C_{A_1,A_2}(z_1,z_2) = \frac{i}{\hbar}\Theta_\gamma(z_1-z_2)\text{Tr}[U_\gamma(-i\hbar\beta,0)U_\gamma(0,z_1)[H_\gamma(z_1),A_1(z_1)]U_\gamma(z_1,z_2)A_2(z_2)U_\gamma(z_2,0)]$$

$$+ \frac{i}{\hbar}\Theta_\gamma(z_2-z_1)\text{Tr}[U_\gamma(-i\hbar\beta,0)U_\gamma(0,z_2)A_2(z_2)U_\gamma(z_2,z_1)[H_\gamma(z_1),A_1(z_1)]U_\gamma(z_1,0)]$$

$$+ \delta(z_1-z_2)\text{Tr}\left\{U_\gamma(-i\hbar\beta,0)U_\gamma(0,z_1)A_1(z_1)U_\gamma(z_1,z_2)A_2(z_2)U_\gamma(z_2,0)\right\}$$

$$- \delta(z_1-z_2)\text{Tr}\left\{U_\gamma(-i\hbar\beta,0)U_\gamma(0,z_2)A_2(z_2)U_\gamma(z_2,z_1)A_1(z_1)U_\gamma(z_1,0)\right\},$$

$$\frac{\partial}{\partial z_2}C_{A_1,A_2}(z_1,z_2) = \frac{i}{\hbar}\Theta_\gamma(z_1-z_2)\text{Tr}[U_\gamma(-i\hbar\beta,0)U_\gamma(0,z_1)A_1(z_1)U_\gamma(z_1,z_2)[H_\gamma(z_2),A_2(z_2)]U_\gamma(z_2,0)]$$

$$+ \frac{i}{\hbar}\Theta_\gamma(z_2-z_1)\text{Tr}[U_\gamma(-i\hbar\beta,0)U_\gamma(0,z_2)[H_\gamma(z_2),A_2(z_2)]U_\gamma(z_2,z_1)A_1(z_1)U_\gamma(z_1,0)]$$

$$- \delta(z_1-z_2)\text{Tr}\left\{U_\gamma(-i\hbar\beta,0)U_\gamma(0,z_1)A_1(z_1)U_\gamma(z_1,z_2)A_2(z_2)U_\gamma(z_2,0)\right\}$$

$$+ \delta(z_1-z_2)\text{Tr}\left\{U_\gamma(-i\hbar\beta,0)U_\gamma(0,z_2)A_2(z_2)U_\gamma(z_2,z_1)A_1(z_1)U_\gamma(z_1,0)\right\}. \tag{77}$$

More general correlation functions can be obtained by including more operators in the trace (74). For instance, we introduce the $n$-operator correlation function as

$$C_{A_1,..A_n}(z_1,...z_n) = \text{Tr}\left[\mathcal{T}_\gamma\left\{e^{-\frac{i}{\hbar}\int_\gamma H_\gamma(z)dz}A_1(z_1)...A_n(z_n)\right\}\right]. \tag{78}$$

In this manuscript we are mainly interested in the case in which the operators $A_i$ are products of bosonic or fermionic creation and annihilation operators. However, the equations of motion (77) can be solved only in specific cases, for instance when $H_\gamma$ is quadratic and $A_{1,2}$ are single creation/annihilation operators. At the contrary, when $H_\gamma$ is not quadratic, the equations (77) are not closed in $G_{A_1,A_2}$ and depend on higher order correlation functions, giving rise to the Martin-Schwinger hierarchy [20].

## B.1 Calculation of the non-interacting Green's functions

Using the results of App. B it is easy to find that for a bosonic operator $a$ governed by the Hamiltonian $H_0 = \omega a^\dagger a$, we have

$$-i\frac{d}{dz'}\langle\mathcal{T}_\gamma\{a(z)a^\dagger(z')\}\rangle = \langle\mathcal{T}_\gamma\{a(z)[\omega a^\dagger(z')a(z'),a^\dagger(z')]\}\rangle + i[a(z),a^\dagger(z')]\delta(z-z')$$

$$= \omega\langle\mathcal{T}_\gamma\{a(z)a^\dagger(z')\}\rangle + i\delta(z-z'), \tag{79}$$

where the $\delta$-term comes from the fact that, if $z' > z$, the ordering operator switches the position of $a$ and $a^\dagger$. Substituting the definition of the Green's function into the equation above, we have for a bosonic Green's function,

$$\frac{d}{dz'}G_b^{(0)}(z,z') = i\omega G_b^{(0)}(z,z') + i\delta(z-z'). \tag{80}$$

Table 1: The values of the contour propagator $G^{(0)}(z_1, z_2)$ for selected values of the vertex time arguments.

|  | $z_2 \in \gamma_+$ | $z_2 \in \gamma_-$ |
|---|---|---|
| $z_1 \in \gamma_+$ | $G^{(0)T}$ | $G^{(0)>}$ |
| $z_1 \in \gamma_-$ | $G^{(0)<}$ | $G^{(0)\bar{T}}$ |

We can derive the same equation for a fermionic operator $c$ as follows,

$$
\begin{aligned}
-i\frac{d}{dz'}\langle \mathcal{T}_\gamma\{c(z)c^\dagger(z')\}\rangle &= \langle \mathcal{T}_\gamma\{c(z)[\omega c^\dagger(z')c(z'), c^\dagger(z')]\}\rangle + i[c(t'), c^\dagger(t')]\delta(z-z') \\
&= \omega\langle \mathcal{T}_\gamma\{c(z)c^\dagger(z')\}\rangle + i\delta(z-z'),
\end{aligned}
\tag{81}
$$

which is formally identical to Eq. (80). The solution of Eq. (80) for a generic GF reads

$$
G_{b,f}^{(0)}(z,z') = -ie^{-i\omega(z-z')}\left\{A\left[\Theta_\gamma(z-z') + \Theta_\gamma(z'-z)\right] + \Theta_\gamma(z-z')\right\}.
\tag{82}
$$

The constant $A$ in the equation above can be determined by means of the boundary conditions [20]: if we replace any of the arguments on the contour with the earliest and latest instants on the contour, the two results should be the same (for bosons) or differ by a sign (for fermions). For a two point GF we thus obtain

$$
\begin{aligned}
G_{b,f}^{(0)}(-i\hbar\beta, t') &= \pm G_{b,f}^{(0)}(0, t'), \\
G_{b,f}^{(0)}(t, -i\hbar\beta) &= \pm G_{b,f}^{(0)}(t, 0),
\end{aligned}
\tag{83}
$$

which leads to Eq. (27) of the main text. When $\lambda = 0$ and $t = t'$, Eqs. (28) represent the components of the non-interacting density matrix [40]. Choosing instead both time arguments on $\gamma_-$ or $\gamma_+$ we obtain the time ordered and anti-time ordered GF

$$
\begin{aligned}
G_{b,f}^{(0)T}(t_-, t'_-) &= -ie^{-i\omega(t-t')}\left[\pm n\Theta(t'-t) + \bar{n}\Theta(t-t')\right], \\
G_{b,f}^{(0)\bar{T}}(t_+, t'_+) &= -ie^{-i\omega(t-t')}\left[\pm n\Theta(t-t') + \bar{n}\Theta(t'-t)\right],
\end{aligned}
\tag{84}
$$

where $\Theta$ is the Heaviside step function on the real axis. These functions coincide with the conventional Green's functions. A summary of the main GF components according to the position of the arguments on the contour is given in Table 1. It is possible to consider more general components by choosing the arguments in the new branches of the contour, e.g., a mixed non-interacting GF $G_{b,f}^{(0)}(t, i\tau)$ for $t \in \gamma_\pm$ and $i\tau \in \gamma_\uparrow$. These new GFs play no role in discussing the work statistics in the switching on/off scenario (19), but are relevant when considering the more general assumption (20). We will discuss them more in detail in App. D.

## C  Examples

Let us consider a simple Holstein coupling model, in which a fermionic energy level is shifted by an amount that depends on the position of a quantum harmonic oscillator. The Hamiltonian of this model is given by

$$
H = \hbar\omega_f c^\dagger c + \hbar\omega_b a^\dagger a + \hbar\chi\omega_\chi c^\dagger c(a + a^\dagger).
\tag{85}
$$

The Feynman diagram contributing to the second-order expansion of Eq. (29) for this potential is represented in Fig. 3A. The only other connected diagram contributing at second order is

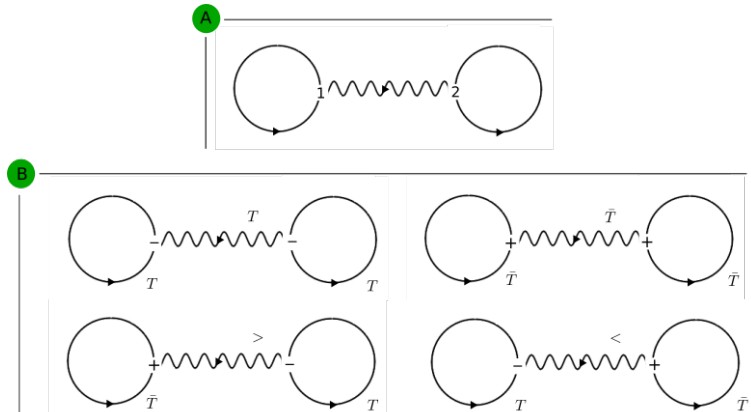

Figure 6: Dumbbell diagrams appearing in the calculations of the work statistics for the Hamiltonian (85).

represented in Fig. 6A, with the associated diagrams for the components given in Fig. 6B. We thus have to take in account a total of 8 diagrams, which we label as $d = 1, \ldots, 4$ for the diagrams in Fig. 3B and $d = 5, \ldots, 8$ for the diagrams in Fig. 6. The diagrams $d = 1, 2, 5, 6$ depend only on time-ordered and anti-time-ordered Green's functions, that according to the definition of $E_d$ given in Sec. 4.2 have $E_d = 0$. Therefore, by inspecting the expression for the cumulant generating function in (33) we realize that these diagrams give a vanishing contribution since $(e^{\lambda E_d} - 1) = 0$.

At the contrary, the three propagators in the third and fourth diagrams ($d = 3, 4$) in Fig. 3 give a non-zero contribution to the cumulant generating function. Starting directly from Eq. (32), and using the notation $G_{b,f}^{(0)}(t_1, t_2)|_{\lambda=0}$ to indicate a GF in which $\lambda$ has been set to 0, we end up with

$$C_W^{(2)}(\lambda, t_f)_{d=3,4} \tag{86}$$

$$= i\chi^2 \omega_\chi^2 \int_0^{t_f} \int_0^{t_f} dt_1 dt_2 G_f^{(0)<}(t_1, t_2) G_f^{(0)>}(t_2, t_1) [G_b^{(0)>}(t_1, t_2) - G_b^{(0)>}(t_1, t_2)|_{\lambda=0}]$$

$$+ i\chi^2 \omega_\chi^2 \int_0^{t_f} \int_0^{t_f} dt_1 dt_2 G_f^{(0)<}(t_1, t_2) G_f^{(0)>}(t_2, t_1) [G_b^{(0)<}(t_1, t_2) - G_b^{(0)<}(t_1, t_2)|_{\lambda=0}],$$

with a prefactor $-\frac{1}{2} \times 2 \times -1 \times i^3 \times -1 = i$ in which $-\frac{1}{2}$ appears in every second order diagram, 2 comes from the freedom of exchanging the two vertices in fig 3, $-1$ from the single $+$ in Fig. 3, $i^3$ from the definition of many body Green's function (see Eqs. (25) and (30)) and $-1$ is the factor prescribed by the Wick theorem (see for instance [20]). The energy jumps for these diagrams $d = 3, 4$ are $E_3 = -\hbar\omega_b$ and $E_4 = \hbar\omega_b$, respectively, as we can verify directly by replacing the components in Eq. (86),

$$C_W^{(2)}(\lambda, t_f)_{d=3,4} = i\chi^2 \omega_\chi^2 \iint_0^{t_f} -i(1 - n_f(\omega_f))[in_f(\omega_f)][-i(1 + n_b(\omega_b))e^{\hbar\omega_b\lambda}e^{-i\omega_b(t_1-t_2)}] dt_1 dt_2 \tag{87}$$

$$+ i\chi^2 \omega_\chi^2 \iint_0^{t_f} -i(1 - n_f(\omega_f))[in_f(\omega_f)][-in_b(\omega_b)e^{-\hbar\omega_b\lambda}e^{i\omega_b(t_1-t_2)}] dt_1 dt_2$$

$$-i\chi^2\omega_\chi^2\iint_0^{t_f}[-i(1-n_f(\omega_f))[in_f(\omega_f)][-i(1+n_b(\omega_b))e^{-i\omega_b(t_1-t_2)}-in_b(\omega_b)e^{i\omega_b(t_1-t_2)}]dt_1dt_2$$

$$=\frac{2\chi^2\omega_\chi^2}{\omega_b^2}(1-\cos(\omega_bt_f))[n_f(\omega_f)-n_f(\omega_f)^2][(1+n_b(\omega_b))(e^{\hbar\omega_b\lambda}-1)+n_b(e^{-\hbar\omega_b\lambda}-1))].$$

By direct comparison with Eq. (33), we find

$$\Gamma_3(t_f)=\gamma_{\omega_b}(t_f)[n_f(\omega_f)-n_f^2(\omega_f)][1+n_b(\omega_b)],$$
$$\Gamma_4(t_f)=\gamma_{\omega_b}(t_f)[n_f(\omega_f)-n_f^2(\omega_f)]n_b(\omega_b),\tag{88}$$

with $\gamma_{\omega_b}(t_f)=\frac{2\chi^2\omega_\chi^2}{\omega_b^2}[1-\cos(\omega_bt_f)]$. The contributions of the diagrams $d=7,8$ represented in Fig. 6 are the same as the ones computed for $d=3,4$ a part from the prefactor $(-n_f^2+n_f)$ being exchanged with $n_f^2$. Summing all the contributions, we obtain

$$C_W^{(2)}(\lambda,t_f)=\sum_\pm\gamma_{\omega_b}(t_f)n_f(\omega_f)n_b(\mp\omega_b)(e^{\pm\hbar\omega_b\lambda}-1).\tag{89}$$

In App. E we compute the generating function of the work exactly, and check that the expansion up to the second order in $\chi$ indeed leads back to Eq. (89).

Another simple application is the anharmonic oscillator with Hamiltonian

$$H=\hbar\omega_b a^\dagger a+\hbar\chi\omega_\chi(a^{\dagger 2}+a^2).\tag{90}$$

As usual, we consider the system as initialized in a Gibbs state of the unperturbed Hamiltonian, switch on the anharmonic term at time zero and perform energy measurements at times 0 and $t$, before switching on and after switching off the perturbation. The vertex in Eq. (90) is given by two inward or outward lines, that at second order produces a single diagram (a loop with two bosonic propagators between time $t_1$ and $t_2$). The contribution of such a diagram gives the cumulant generating function at second order

$$C_W^{(2)}(\lambda,t_f)=2\gamma_{2\omega_b}(t_f)[(e^{-2\hbar\lambda\omega_b}-1)n_b^2+(1+n_b)^2(e^{2\hbar\lambda\omega_b}-1)].\tag{91}$$

In the notation of Eq. (33), this cumulant generating function consists of two rescaled Poissonian energy jumps with energies equal to $E_d=\pm2\hbar\omega_b$ and rates given by $\gamma_{2\omega_b}(t_f)n_b^2$ and $\gamma_{2\omega_b}(t_f)(1+n_b)^2$, respectively.

# D  The role of the other GF components on the contour

When writing the contour GF in its component we also need to consider situations in which one of the arguments is in the vertical track $\gamma_\uparrow$. We can define a non-interacting GF with both arguments on the vertical track, $G_{b,f}^{(0)\uparrow\uparrow}(r,r')\equiv G_{b,f}^{(0)}(ir,ir')$ with $ir,ir'\in\gamma_\uparrow$. As a consequence we have

$$G_{b,f}^{(0)\uparrow\uparrow}(r,r')=-ie^{\omega(r-r')}\big[\pm n\Theta(r'-r)+\bar{n}\Theta(r-r')\big].\tag{92}$$

Another possibility is represented by the case in which one argument is on the horizontal branches and one on $\gamma_\uparrow$. We can identify four different cases $G_{b,f}^{(0)\uparrow+}(r,t)=G_{b,f}^{(0)}(ir,t)$, $G^{(0)\uparrow-}(r,t)=G_{b,f}^{(0)}(ir,t)$, $G_{b,f}^{(0)+\uparrow}(t,r)=G_{b,f}^{(0)}(t,ir)$, $G_{b,f}^{(0)-\uparrow}(t,r)=G_{b,f}^{(0)}(t,ir)$.

We explicitly write this four new components below

$$G_{b,f}^{(0)\uparrow +}(r,t') = \mp i n e^{\omega(r-\lambda)+i\omega(t_f-t')}; \qquad G_{b,f}^{(0)\uparrow -}(r,t') = -i\bar{n}e^{\omega r + i\omega(t'-t_f)};$$
$$G_{b,f}^{(0)+\uparrow}(t,r') = -i\bar{n}e^{-\omega(r-\lambda)-i\omega(t_f-t')}; \quad G_{b,f}^{(0)-\uparrow}(t,r') = \mp i n e^{-\omega r - i\omega(t'-t_f)}. \tag{93}$$

This definitions pave the way to an application of the theory to protocols beyond the switching on/off paradigm in which $\chi(z)$ is defined as in Eq. (20). For a generic integral of a function in the modified contour we have

$$\int_{\gamma} \chi(z)f(z)dz = \int_{\gamma_+} \chi(z)f(z)dz + \int_{\gamma_-} \chi(z)f(z)dz + \int_{\gamma_\uparrow} \chi(z)f(z)dz$$
$$= \int_0^{t_f} \chi(t)f_-(t)dt - \int_0^{t_f} \chi(t)f_+(t)dt + i\chi(t_f)\int_0^{\hbar\lambda} f_\uparrow(r)dr. \tag{94}$$

where $f_\uparrow, f_\pm$ are the components of the function $f$ in the modified contour. Notice that the contribution of the downward and Matsubara tracks is absent, since we are implicitly assuming $\chi(z) = 0$ on $\gamma_{M,\downarrow}$ as prescribed by the protocols (19) and (20). The same decomposition of the integral can be done for the equations arising from the perturbative expansion, like Eq. (30).

To make the calculations simpler, let us carry them on in the case in which the switching function is given by

$$\chi(z) = \begin{cases} 0 & \text{for } z \in \gamma_K, \\ 0 & \text{for } z \in \gamma_{M,\downarrow}, \\ \chi & \text{for } z \in \gamma_\uparrow, \end{cases} \tag{95}$$

this case correspond to a system in which the Hamiltonian $\omega_\chi H_1$ is quenched and then immediately measured (without leaving the system enough time to evolve). We can compute the generating function for the case of the Hamiltonian (85) discussed in the main text. The diagrams involved are the same (see A panels of Fig. 3 and 6) but we integrate over the vertical branch, and the only Green's function appearing in the calculations is $G_{b,f}^{(0)\uparrow\uparrow}$. The integrals in Eq. (30) reduce to

$$C_W^{(2)}(\lambda, t_f)_{d=1} = i\chi^2\omega_\chi^2 \int_0^\lambda \int_0^\lambda dr_1 dr_2 G_b^{(0)\uparrow\uparrow}(r_1,r_2) G_f^{(0)\uparrow\uparrow}(r_1,r_2) G_f^{(0)\uparrow\uparrow}(r_2,r_1)$$
$$= -\chi^2\omega_\chi^2 \int_0^\lambda \int_0^\lambda dr_1 dr_2 (-n_f\bar{n}_f)e^{\omega_b(r_1-r_2)}[n_b\Theta(r_2-r_1) + \bar{n}_b\Theta(r_1-r_2)]$$
$$= \chi^2\omega_\chi^2 n_f\bar{n}_f [n_b \frac{\hbar\lambda\omega_b + e^{-\hbar\lambda\omega_b} - 1}{\omega_b^2} + \bar{n}_b \frac{-\hbar\lambda\omega_b + e^{\hbar\lambda\omega_b} - 1}{\omega_b^2}]$$
$$= \chi^2\omega_\chi^2 \frac{n_f - n_f^2}{\omega_b^2} [n_b(e^{-\hbar\lambda\omega_b} - 1) + (1+n_b)(e^{\hbar\lambda\omega_b} - 1) - \hbar\lambda\omega_b]. \tag{96}$$

Where we denoted with $d = 1$ the contribution of the diagram in Fig.3A with the value of the vertex variables to be chosen in $\gamma_\uparrow$. The contribution of the dumbbell diagram is instead given by

$$
\begin{aligned}
M_W^{(2)}(\lambda, t_f)_{d=2} &= -i\chi^2\omega_\chi^2 \int_0^\lambda \int_0^\lambda dr_1 dr_2 G_b^{(0)\uparrow\uparrow}(r_1, r_2) G_f^{(0)\uparrow\uparrow}(r_1, r_1^+) G_f^{(0)\uparrow\uparrow}(r_2, r_2^+) \\
&= \chi^2\omega_\chi^2 \int_0^\lambda \int_0^\lambda dr_1 dr_2 n_f^2 e^{\omega_b(r_1-r_2)}[n_b\Theta(r_2-r_1) + \bar{n}_b\Theta(r_1-r_2)] \\
&= \chi^2\omega_\chi^2 n_f^2 [n_b \frac{\hbar\lambda\omega_b + e^{-\hbar\lambda\omega_b} - 1}{\omega_a^2} + \bar{n}_b \frac{-\hbar\lambda\omega_b + e^{\hbar\lambda\omega_b} - 1}{\omega_b^2}] \\
&= \frac{\chi^2\omega_\chi^2 n_f^2}{\omega_b^2} [n_b(e^{-\hbar\lambda\omega_b} - 1) + (1+n_b)(e^{\hbar\lambda\omega_b} - 1) - \hbar\lambda\omega_b].
\end{aligned}
\tag{97}
$$

The cumulant generating function at second order can be obtained by summing the two contributions in (96) and (97).

# E  Work statistics in the dispersive coupling case: non-perturbative approach

The dispersive coupling Hamiltonian is common to several models in open quantum systems,

$$
H = \hbar\omega_b a^\dagger a + \hbar\omega_f c^\dagger c + \hbar\chi\omega_\chi c^\dagger c(a^\dagger + a)
\tag{98}
$$

where $a$ and $c$ denote a bosonic and fermionic annihilation operator, respectively. The fermionic operators in the Hamiltonian (98) can be expressed as Pauli matrices acting over a two dimensional Hilbert space, by defining $\sigma^+ = c^\dagger$, $\sigma^z = 2c^\dagger c - 1$. The anticommutation relations $\{\sigma^-, \sigma^+\} = \{c, c^+\} = 1$ are preserved and we obtain

$$
H = \hbar\omega_b a^\dagger a + \hbar\left(\frac{\sigma_z}{2} + \frac{1}{2}\right)(\omega_f + \chi\omega_\chi a^\dagger + \chi\omega_\chi a).
\tag{99}
$$

The Hamiltonian commutes with $\sigma_z$, so we can write the time evolution operator as

$$
U(t,0) = \mathcal{T}\{e^{-\frac{i}{\hbar}\int_0^t H(s)ds}\} = \begin{pmatrix} e^{-\frac{i}{\hbar}(\hbar\omega_b a^\dagger a + \hbar\omega_f + \hbar\chi\omega_\chi a^\dagger + \hbar\chi\omega_\chi a)t} & 0 \\ 0 & e^{-i\omega_b a^\dagger a t} \end{pmatrix}.
\tag{100}
$$

The top-left element of the matrix (100) is the same as from a driven harmonic oscillator. We can define $H_b = \hbar\omega_b(a^\dagger a + 1/2)$ and

$$
\langle 1|e^{-\frac{i}{\hbar}Ht}|1\rangle = e^{-i(\omega_f-\omega_b)t} e^{-\frac{i}{\hbar}H_b t - i\chi\omega_\chi(a+a^\dagger)t},
\tag{101}
$$

the exponential above can be written in terms of displacement operators [78]

$$
e^{-\frac{i}{\hbar}H_b t - i\chi\omega_\chi(a+a^\dagger)t} = e^{\frac{i}{\hbar}\theta(t)} D[\delta(t)] e^{-\frac{i}{\hbar}H_b t},
\tag{102}
$$

where $\theta(t)$ is a phase factor and

$$
D[\delta(t)] = \exp\{(\delta(t)a^\dagger - \delta^*(t)a)\},
\tag{103}
$$

is the displacement operator of argument $\delta(t)$. The time dependent parameter $\delta(t)$ is connected to the classical solution for the Hamilton equations for the variable $\alpha(t) = \frac{x(t)+ip(t)}{\sqrt{2}}$, with $\alpha(0) = 0$. This variable evolve as $\alpha(t) = \alpha e^{-i\omega t} + \delta(t)$, where $\delta(t) = \frac{\chi\omega_\chi}{\omega_b}(1 - e^{-i\omega t})$. For

counting the work statistics in the sudden quench scenario we have the following generating function

$$M_W(\lambda, t_f) = \frac{1}{Z}\text{Tr}[e^{\lambda H_0}U(t_f, 0)e^{-(\lambda+\beta)H_0}U^\dagger(t_f, 0)], \tag{104}$$

so that it is clear using Eq. (100) that we have to evaluate the tilted displacement $D[\delta(t_f), \lambda] = e^{\lambda H_0}D[\delta(t_f)]e^{-\lambda H_0}$. Since $e^{\lambda H_0}e^{-\lambda H_0} = 1$ we can bring the two matrices at the exponent in the displacement operator and obtain

$$D[\delta(t_f), \lambda] = \exp\left[\delta(t_f)e^{\lambda H_0}a^\dagger e^{-\lambda H_0} - \delta^*(t_f)e^{\lambda H_0}ae^{-\lambda H_0}\right]$$
$$= \exp\left[\delta(t_f)e^{\hbar\lambda\omega_b}a^\dagger - \delta^*(t_f)e^{-\hbar\lambda\omega_b}a\right]. \tag{105}$$

Using Eq. (104)) we find

$$M_W(\lambda, t_f) = \frac{1}{Z}\text{Tr}[U^\dagger(t_f, 0)U_\lambda(t_f, 0)\begin{pmatrix} e^{-\hbar\beta\omega_b a^\dagger a}e^{-\hbar\beta\omega_f} & 0 \\ 0 & e^{-\hbar\beta\omega_b a^\dagger a} \end{pmatrix}]$$

$$= \frac{1}{Z}\text{Tr}[\begin{pmatrix} D[-\delta(t_f)]D[\delta(t_f), \lambda] & 0 \\ 0 & 1 \end{pmatrix}\begin{pmatrix} e^{-\hbar\beta\omega_b a^\dagger a}e^{-\hbar\beta\omega_f} & 0 \\ 0 & e^{-\hbar\beta\omega_b a^\dagger a} \end{pmatrix}], \tag{106}$$

We are interested in a weak perturbation expansion of equation above, that is $\delta(t_f) << \omega_b$, so we can Taylor expand $D[-\delta(t_f)]D[\delta(t_f), \lambda]$ keeping only the terms with the same number of $a$ and $a^\dagger$ operators, the others averaging to 0:

$$D[-\delta(t_f)]D[\delta(t_f), \lambda] = 1 + |\delta(t_f)|^2[(e^{\hbar\lambda\omega_b} - 1)aa^\dagger + (e^{-\hbar\omega_b\lambda} - 1)a^\dagger a] + O(\chi^3). \tag{107}$$

The final expression for $M(\lambda, t)$ thus reads

$$M_W(\lambda, t_f) = \frac{1}{Z_f}(1 + e^{-\hbar\beta\omega_f}\{1 + |\delta(t_f)|^2[(e^{\hbar\lambda\omega_b} - 1)(1 + n_b) + (e^{-\hbar\omega_b\lambda} - 1)n_b]\}) \tag{108}$$

$$= 1 + 4\frac{\chi^2\omega_\chi^2}{\omega_b^2}\sin^2\left(\frac{\omega_b t_f}{2}\right)n_f[(e^{\hbar\lambda\omega_b} - 1)(1 + n_b) + (e^{-\hbar\omega_b\lambda} - 1)n_b], \tag{109}$$

computing the logarithm to find the expansion for the cumulant generating function and noting that $2\sin^2(\frac{\omega_b t_f}{2}) = 1 - \cos(\omega_b t_f)$ we obtain and confirm the result of the Eq. (89).

## F Summation over the momentum variables in the path integral

To write the path integral in terms of the position coordinates we consider the quantity $\mathcal{S} = i/\hbar S$ in the exponent of the Eq. (39). From the main text we have

$$\mathcal{S} = \frac{i}{\hbar}\int_\gamma \{\frac{d}{dz}x(z)p(z) - \frac{p(z)^2}{2m} - V[x(z)]\}dz, \tag{110}$$

where for ease of notation we exclude the parameter $\alpha$ from the argument of $V$. Depending on the branch we are considering the integration of the momentum variables leads to different results. Adopting discrete notation, the exponent of the path integral reads

$$\text{on } \gamma_- \rightarrow -\frac{i}{\hbar}\frac{p^2(t)}{2m}\Delta t + \frac{i}{\hbar}\frac{dx(t)}{dt}p(t)\Delta t; \quad \text{on } \gamma_+ \rightarrow \frac{i}{\hbar}\frac{p^2(t)}{2m}\Delta t - \frac{i}{\hbar}\frac{dx(t)}{dt}p(t)\Delta t; \tag{111}$$

$$\text{on } \gamma_\uparrow \rightarrow \frac{p_\uparrow^2(\tau)}{2m}\Delta\tau - i\frac{dx_\uparrow(\tau)}{d\tau}p_\uparrow(\tau)\Delta\tau; \quad \text{on } \gamma_\downarrow, \gamma_M \rightarrow -\frac{p_{\downarrow,M}^2(\tau)}{2m}\Delta\tau + i\frac{dx_{\downarrow,M}(\tau)}{d\tau}p(\tau)\Delta\tau,$$

where we have assumed $z = t + i\tau$. After eliminating the momentum variables from the action of the path integral via gaussian integration, the new exponents on the different branches of the contour read

$$
\begin{aligned}
&\text{on } \gamma_- \;\to\; \frac{i}{2\hbar} m \left[ \frac{dx(t)}{dt} \right]^2 \Delta t; \quad \text{on } \gamma_+ \;\to\; -\frac{i}{2\hbar} m \left[ \frac{dx(t)}{dt} \right]^2 \Delta t; \\
&\text{on } \gamma_\uparrow \;\to\; \frac{1}{2} m \left[ \frac{dx(\tau)}{d\tau} \right]^2 \Delta\tau; \quad \text{on } \gamma_\downarrow, \gamma_M \;\to\; -\frac{1}{2} m \left[ \frac{dx(\tau)}{d\tau} \right]^2 \Delta\tau.
\end{aligned}
\tag{112}
$$

The terms above give the contribution to exponent in the path integral and can be written respectively like $iT, -iT, T, -T$ with $T$ the kinetic energy. After going back to the contour variable $z$ this kinetic contribution sums with the potential energy and gives

$$
\begin{aligned}
\text{on } \gamma_- &\;\to\; (iT - iV)\Delta t = (iT - iV)\Delta z = i\mathcal{L}\,dz; \\
\text{on } \gamma_+ &\;\to\; (-iT + iV)\Delta t = (-iT + iV)(-\Delta z) = i\mathcal{L}\,dz; \\
\text{on } \gamma_\uparrow &\;\to\; (T + V)\Delta\tau = -i(T + V)\Delta z = i\mathcal{L}_\uparrow\,dz; \\
\text{on } \gamma_\downarrow, \gamma_M &\;\to\; -(T + V)\Delta\tau = i(T + V)(-\Delta z) = i\mathcal{L}_{\downarrow/M}\,dz.
\end{aligned}
\tag{113}
$$

In total we can write the exponent of the path integral as

$$
S = \frac{i}{\hbar} \int_\gamma \mathcal{L}_\gamma \left[ x(z), x'(z) \right] dz,
\tag{114}
$$

where $\mathcal{L}$ is defined on the contour as

$$
\mathcal{L}_\gamma(z) =
\begin{cases}
\mathcal{L}[x(t), \dot{x}(t)] & \text{for } z \in \gamma_K, \\
\mathcal{L}_\uparrow, \mathcal{L}_\downarrow, \mathcal{L}_M & \text{for } z \in \gamma_{\uparrow,\downarrow,M}.
\end{cases}
\tag{115}
$$

## G Semiclassical limit

### G.1 Hamiltonian action in the modified contour

The same manipulations done to obtain (46) can also be done before the integration of the momentum variables, at the level of the action (40):

$$
\begin{aligned}
\frac{i}{\hbar} S = &\; \frac{i}{\hbar} \int_\gamma dz \left\{ \frac{dx(z)}{dz} p(z) - H_\gamma[x(z), p(z)] \right\} \\
= &\; \frac{i}{\hbar} \int_{\gamma_\ominus} dz \left\{ \frac{dx_\ominus(z)}{dz} p_\ominus(z) - H_\gamma[x_\ominus(z), p_\ominus(z)] \right\} + \frac{i}{\hbar} \int_{\gamma_\oplus} dz \left\{ \frac{dx_\oplus(z)}{dz} p_\oplus(z) - H_\gamma[x_\oplus(z), p_\oplus(z)] \right\} \\
= &\; \frac{i}{\hbar} \int_{\gamma_\ominus} dz \left\{ \frac{dx_\ominus(z)}{dz} p_\ominus(z) - H_\gamma[x_\ominus(z), p_\ominus(z)] \right\} - \frac{i}{\hbar} \int_{\gamma_\ominus} dz^* \left\{ \frac{dx_\oplus(z^*)}{dz^*} p_\oplus(z^*) - H_\gamma[x_\oplus(z^*), p_\oplus(z^*)] \right\} \\
= &\; \frac{i}{\hbar} \int_{\gamma_\ominus} d\,\mathrm{Re}\,z \left\{ \frac{dx_\ominus(z)}{dz} p_\ominus(z) - \frac{dx_\oplus(z^*)}{dz} p_\oplus(z^*) - \left\{ H_\gamma[x_\ominus(z), p_\ominus(z)] - H_\gamma[x_\oplus(z^*), p_\oplus(z^*)] \right\} \right\} \\
&\; - \frac{1}{\hbar} \int_{\gamma_\ominus} d\,\mathrm{Im}\,z \left\{ \frac{dx_\ominus(z)}{dz} p_\ominus(z) - \frac{dx_\oplus(z^*)}{dz} p_\oplus(z^*) - \left\{ H_\gamma[x_\ominus(z), p_\ominus(z)] + H_\gamma[x_\oplus(z^*), p_\oplus(z^*)] \right\} \right\},
\end{aligned}
\tag{116}
$$

where we used that in the vertical branches, in which $d\,\mathrm{Im}\,z \neq 0$, we have $\frac{d}{dz^*} = -\frac{d}{dz}$, while in the horizotontal branches, in which $d\,\mathrm{Re}\,z \neq 0$, we have $\frac{d}{dz^*} = \frac{d}{dz}$. To have an expression of the action in terms of the quantum and classical components of the fields we apply the linear

transformation (47) and substitute $x_{cl}(z) = 1/2[x_{\ominus}(z)+x_{\oplus}(z^*)]$, $x_q(z) = 1/2[x_{\ominus}(z)-x_{\oplus}(z^*)]$, $p_q(z) = 1/2[p_{\ominus}(z)-p_{\oplus}(z^*)]$ and $p_{cl}(z) = 1/2[p_{\ominus}(z)+p_{\oplus}(z^*)]$, therefore

$$\frac{dx_{\ominus}(z)}{dz}p_{\ominus}(z) - \frac{dx_{\oplus}(z^*)}{dz}p_{\oplus}(z^*) = \frac{dx_{cl}(z)}{dz}[p_{\ominus}(z)-p_{\oplus}(z^*)] + \frac{dx_q(z)}{dz}[p_{\ominus}(z)+p_{\oplus}(z^*)]. \quad (117)$$

From it, using again Eq. (47), we obtain an expression only in terms of the quantum and classical components

$$\frac{dx_{\ominus}(z)}{dz}p_{\ominus}(z) - \frac{dx_{\oplus}(z^*)}{dz}p_{\oplus}(z^*) = 2\frac{dx_{cl}(z)}{dz}p_q(z) + 2\frac{dx_q(z)}{dz}p_{cl}(z). \quad (118)$$

If we replace the equation above in Eq. (116) we obtain the action in terms of the quantum and classical variables

$$\frac{i}{\hbar}S = \frac{i}{\hbar}\int_{\gamma_{\ominus}} d\,\mathrm{Re}z\left\{2\frac{dx_{cl}(z)}{dz}p_q(z) + 2\frac{dx_q(z)}{dz}p_{cl}(z)\right\} \quad (119)$$

$$- \frac{i}{\hbar}\int_{\gamma_{\ominus}} d\,\mathrm{Re}z\left\{H_{\gamma}[x_{cl}(z)+x_q(z),p_{cl}(z)+p_q(z)] - H_{\gamma}[x_{cl}(z)-x_q(z),p_{cl}(z)-p_q(z)]\right\}$$

$$- \frac{1}{\hbar}\int_{\gamma_{\ominus}} d\,\mathrm{Im}z\left\{2\frac{dx_{cl}(z)}{dz}p_q(z) + 2\frac{dx_q(z)}{dz}p_{cl}(z)\right\}$$

$$+ \frac{1}{\hbar}\int_{\gamma_{\ominus}} d\,\mathrm{Im}z\left\{H_{\gamma}[x_{cl}(z)+x_q(z),p_{cl}(z)+p_q(z)] + H_{\gamma}[x_{cl}(z)-x_q(z),p_{cl}(z)-p_q(z)]\right\}.$$

## G.2 Expansion of the action with respect to the fields

In this section, we perform the expansion of the action (119) under the assumptions that the quantum fields admit an expansion in powers of $\hbar$. Now we can expand both the integrals over the quantum variables $x_q$ and $p_q$ up to the second order thus for the horizontal track we will have

$$H_{\gamma}[x_{cl}(z)+x_q(z),p_{cl}(z)+p_q(z)] - H_{\gamma}[x_{cl}(z)-x_q(z),p_{cl}(z)-p_q(z)]$$
$$= 2\frac{p_{cl}p_q}{m} + V[x_{cl}(z)+x_q(z)] - V[x_{cl}(z)-x_q(z)] \approx 2\frac{p_{cl}p_q}{m} + 2x_q\frac{\partial}{\partial x_{cl}}V[x_{cl}], \quad (120)$$

and for the vertical branches

$$H_{\gamma}[x_{cl}(z)+x_q(z),p_{cl}(z)+p_q(z)] + H_{\gamma}[x_{cl}(z)-x_q(z),p_{cl}(z)-p_q(z)]$$
$$= 2H_{\gamma}[x_{cl}(z),p_{cl}(z)] + \frac{p_q^2(z)}{m} + x_q^2(z)\frac{\partial^2}{\partial x_{cl}^2}V[x_{cl}]. \quad (121)$$

After integrating by parts Eq. (119) and replacing the expansions (120) and (121) we obtain

$$\frac{i}{\hbar}S = \frac{2i}{\hbar}\int_{\gamma_{\ominus}} d\,\mathrm{Re}z\left\{\frac{dx_{cl}(z)}{dz}p_q(z) - \frac{dp_{cl}(z)}{dz}x_q(z) - \left[\frac{p_q p_{cl}}{m} + x_q\frac{\partial}{\partial x_{cl}}V[x_{cl}]\right]\right\} \quad (122)$$

$$- \frac{2}{\hbar}\int_{\gamma_{\ominus}} d\,\mathrm{Im}z\left\{\frac{dx_{cl}(z)}{dz}p_q(z) - \frac{dp_{cl}(z)}{dz}x_q(z) - \left[H_{\gamma}[x_{cl}(z),p_{cl}(z)] + \frac{p_q^2(z)}{2m} + \frac{1}{2}x_q^2(z)\frac{\partial^2}{\partial x_{cl}^2}V[x_{cl}]\right]\right\}.$$

Notice that the boundary terms of the integration by parts vanish as an effect of the boundary conditions $x_q(t_f) = x_q(i\hbar\beta/2) = 0$. Let us decompose the integral over the vertical lines

$(\int_{\gamma_\ominus} d\,\mathrm{Im}\,z)$ in two separate contributions given by the vertical tracks $\gamma_{(\ominus,\uparrow)}$ and $\gamma_{(\ominus,\downarrow),(\ominus,M)}$ where we write the components of the quantum and classical fields as

$$x_{cl/q}(z) = \begin{cases} x_{cl/q\uparrow}(\tau) & \text{for } z = t_f + i\tau \in \gamma_{(\ominus,\uparrow)}, \\ x_{cl/q\downarrow}(\tau) & \text{for } z = i\tau \in \gamma_{(\ominus,\downarrow),(\ominus,M)}. \end{cases} \tag{123}$$

Note that in (41), we identified the components of the branches $\gamma_M$ and $\gamma_\downarrow$ by $x_M(\tau)$ and $x_\downarrow(\tau)$, respectively. However, here on the symmetrized contour, for ease of calculations, we have denoted the fields on the branches $\gamma_{(\ominus,\downarrow)}$ and $\gamma_{(\ominus,M)}$ by $x_{cl/q\downarrow}(\tau)$. In this way we will have

$$-\frac{2}{\hbar}\int_{\gamma_\ominus} d\,\mathrm{Im}\,z\left\{\frac{dx_{cl}(z)}{dz}p_q(z) - \frac{dp_{cl}(z)}{dz}x_q(z) - \left[H_\gamma[x_{cl}(z), p_{cl}(z)] + \frac{p_q^2(z)}{2m} + \frac{1}{2}x_q^2(z)\frac{\partial^2}{\partial x_{cl}^2}V[x_{cl}]\right]\right\} = \tag{124}$$

$$-\frac{2}{\hbar}\int_{-\lambda\hbar/2}^0 d\tau\left\{i\frac{dp_{cl\uparrow}(\tau)}{d\tau}x_{q\uparrow}(\tau) - i\frac{dx_{cl\uparrow}(\tau)}{d\tau}p_{q\uparrow}(\tau) - \left[\frac{p_{cl\uparrow}^2(\tau)}{2m} + V[x_{cl\uparrow}(\tau)] + \frac{p_{q\uparrow}^2(\tau)}{2m} + \frac{1}{2}x_{q\uparrow}^2(\tau)V''[x_{cl\uparrow}(\tau)]\right]\right\}$$

$$+\frac{2}{\hbar}\int_{-\lambda\hbar/2}^{\beta\hbar/2} d\tau\left\{i\frac{dp_{cl\downarrow}(\tau)}{d\tau}x_{q\downarrow}(\tau) - i\frac{dx_{cl\downarrow}(\tau)}{d\tau}p_{q\downarrow}(\tau) - \left[\frac{p_{cl\downarrow}^2(\tau)}{2m} + V[x_{cl\downarrow}(\tau)] + \frac{p_{q\downarrow}^2(\tau)}{2m} + \frac{1}{2}x_{q\downarrow}^2(\tau)V''[x_{cl\downarrow}(\tau)]\right]\right\}.$$

### G.3 Gaussian integration of the fields

In the Eq. (124) the action is quadratic in the fields $x_q, p_q$. They can be eliminated through a Gaussian integration. If define $\frac{i}{\hbar}\tilde{S}$ as the exponent of the path integral after such Gaussian integration, we have

$$\frac{i\tilde{S}}{\hbar} = \frac{1}{\hbar}\int_{-\hbar\lambda/2}^0 \frac{d\tau}{V''[x_{cl\uparrow}(\tau)]}\left[\frac{dp_{cl\uparrow}(\tau)}{d\tau}\right]^2 + \frac{1}{\hbar}\int_{-\hbar\lambda/2}^0 d\tau\, m\left[\frac{dx_{cl\uparrow}(\tau)}{d\tau}\right]^2$$

$$+\frac{2}{\hbar}\int_{-\lambda\hbar/2}^0 d\tau\left\{\frac{p_{cl\uparrow}^2(\tau)}{2m} + V[x_{cl\uparrow}(\tau)]\right\} - \frac{1}{\hbar}\int_{-\hbar\lambda/2}^{\beta\hbar/2} \frac{d\tau}{V''[x_{cl\downarrow}(\tau)]}\left[\frac{dp_{cl\downarrow}(\tau)}{d\tau}\right]^2$$

$$-\frac{1}{\hbar}\int_{-\hbar\lambda/2}^{\beta\hbar/2} d\tau\, m\left[\frac{dx_{cl\downarrow}(\tau)}{d\tau}\right]^2 - \frac{2}{\hbar}\int_{-\lambda\hbar/2}^{\beta\hbar/2} d\tau\left\{\frac{p_{cl\downarrow}^2(\tau)}{2m} + V[x_{cl\downarrow}(\tau)]\right\}. \tag{125}$$

Notice that a byproduct of the path integration above is a change of the normalization factor proportional to $1/\sqrt{V''[x_{cl\downarrow}(\tau)]}$, $1/\sqrt{V''[x_{cl\uparrow}(\tau)]}$. To make further manipulations possible we will assume that the quantities $V''[x_{cl\downarrow}(\tau)], V''[x_{cl\uparrow}(\tau)]$ can be respectively replaced with $V''[x_{cl}(0)], V''[x_{cl}(0)]$ and $V''[x_{cl}(t_f)], V''[x_{cl}(t_f)]$ in the kinetic terms of the action, and the normalization. The idea behind this replacement is that the vertical tracks are short in the semiclassical limit, so that it is possible to replace the values of the fields in $\gamma_\uparrow, \gamma_\downarrow$ with the values of the fields at their boundaries. Let us focus now on the integral in the momentum variables. After defining $\tilde{m}(t_f) = 2/V''[x_{cl\uparrow}(t_f)]$ and $\omega(t_f) = \sqrt{V''[x_{cl\uparrow}(t_f)]/m}$ we have

$$\int \mathcal{D}p_{cl\uparrow}\exp\left\{\frac{1}{\hbar}\int_{-\lambda\hbar/2}^0 d\tau\frac{\tilde{m}(t_f)}{2}\left[\frac{dp_{cl\uparrow}(\tau)}{d\tau}\right]^2 + \frac{\tilde{m}(t_f)\omega^2(t_f)p_{cl\uparrow}^2(\tau)}{2}\right\} = \tag{126}$$

$$\sqrt{\frac{\tilde{m}\omega}{2\pi\hbar\sinh(\hbar\omega\lambda/2)}}\exp\left\{\frac{\tilde{m}\omega}{2\hbar\sinh(\hbar\omega\lambda/2)}\left\{\left[p_{cl}^2(t_f) + p_{cl\uparrow}^2(0)\right]\cosh(\hbar\omega\lambda/2) - 2p_{cl}(t_f)p_{cl\uparrow}(0)\right\}\right\}.$$

where for now we restrict our analysis to $\gamma_\uparrow$. To integral over $p_{cl\uparrow}(0)$ reads

$$
\int \mathcal{D}p_{cl\uparrow} \sqrt{\frac{\tilde{m}\omega}{2\pi\hbar\sinh(\hbar\omega\lambda/2)}} \tag{127}
$$
$$
\times \exp\left\{\frac{\tilde{m}\omega}{2\hbar\sinh(\hbar\omega\lambda/2)}\left\{\left[p_{cl}^2(t_f)+p_{cl\uparrow}^2(0)\right]\cosh(\hbar\omega\lambda/2)-2p_{cl}(t_f)p_{cl\uparrow}(0)\right\}\right\}
$$
$$
=\sqrt{\frac{\tilde{m}\omega}{2\pi\hbar\sinh(\hbar\omega\lambda/2)}}\exp\left\{\frac{\tilde{m}\omega}{2\hbar\sinh(\hbar\omega\lambda/2)}p_{cl}^2(t_f)\cosh(\hbar\omega\lambda/2)\right\}
$$
$$
\times\int dp_{cl\uparrow}\exp\left\{\frac{\tilde{m}\omega}{2\hbar\sinh(\hbar\omega\lambda/2)}\left[p_{cl\uparrow}^2(0)\cosh(\hbar\omega\lambda/2)-2p_{cl}(t_f)p_{cl\uparrow}(0)\right]\right\}
$$
$$
=\sqrt{\frac{1}{2\cosh(\hbar\omega\lambda/2)}}\exp\left\{\frac{p_{cl}^2(t_f)\tilde{m}\omega}{2\hbar\sinh(\hbar\omega\lambda/2)}\left[\cosh(\hbar\omega\lambda/2)-\frac{1}{\cosh(\hbar\omega\lambda/2)}\right]\right\}
$$
$$
=\sqrt{\frac{1}{2\cosh(\hbar\omega\lambda/2)}}\exp\left\{\frac{p_{cl}^2(t_f)\tilde{m}\omega}{2\hbar\cosh(\hbar\omega\lambda/2)}\sinh(\hbar\omega\lambda/2)\right\}.
$$

Similarly for the other vertical branch we will have

$$
\int \mathcal{D}p_{cl\downarrow}\exp\left\{\frac{-1}{\hbar}\int_{-\lambda\hbar/2}^{\beta\hbar/2}d\tau\frac{\tilde{m}(t_f)}{2}\left[\frac{dp_{cl\downarrow}(\tau)}{d\tau}\right]^2+\frac{\tilde{m}(t_f)\omega^2(t_f)p_{cl\downarrow}^2(\tau)}{2}\right\} \tag{128}
$$
$$
=\sqrt{\frac{\tilde{m}\omega}{2\pi\hbar\sinh(\hbar\omega(\lambda+\beta)/2)}}\exp\left\{-\frac{\tilde{m}\omega\cosh(\hbar\omega(\lambda+\beta)/2)}{2\hbar\sinh(\hbar\omega(\lambda+\beta)/2)}\left[p_{cl}^2(0)+p_{cl\downarrow}^2(\beta\hbar/2)\right]\right\}
$$
$$
\times\exp\left\{\frac{\tilde{m}\omega}{2\hbar\sinh(\hbar\omega(\lambda+\beta)/2)}\left\{2p_{cl}(0)p_{cl\downarrow}(\beta\hbar/2)\right\}\right\}.
$$

A further integration over $p_{cl\downarrow}(\beta\hbar/2)$ yields

$$
\int \mathcal{D}p_{cl\downarrow}\exp\left\{\frac{-1}{\hbar}\int_{-\lambda\hbar/2}^{\beta\hbar/2}d\tau\frac{\tilde{m}(t_f)}{2}\left[\frac{dp_{cl\downarrow}(\tau)}{d\tau}\right]^2+\frac{\tilde{m}(t_f)\omega^2(t_f)p_{cl\downarrow}^2(\tau)}{2}\right\} \tag{129}
$$
$$
=\sqrt{\frac{1}{2\cosh(\hbar\omega(\lambda+\beta)/2)}}
$$
$$
\times\exp\left\{\frac{-p_{cl}^2(0)\tilde{m}\omega}{2\hbar\sinh(\hbar\omega(\lambda+\beta)/2)}\left[\cosh(\hbar\omega(\lambda+\beta)/2)-\frac{1}{\cosh(\hbar\omega(\lambda+\beta)/2)}\right]\right\}
$$
$$
=\sqrt{\frac{1}{2\cosh(\hbar\omega(\lambda+\beta)/2)}}\exp\left\{\frac{-p_{cl}^2(0)\tilde{m}\omega}{2\hbar\cosh(\hbar\omega(\lambda+\beta)/2)}\sinh(\hbar\omega(\lambda+\beta)/2)\right\}.
$$

Thus we can write the contribution to the path integral due to vertical lines as

$$
\int Dx_q Dp_q D_{P_{cl}}e^{\frac{1}{\hbar}\int_{(\gamma_\ominus,\downarrow),(\ominus,M)}\Sigma[x_{cl},x_q]d\,\mathrm{Im}\,z}e^{\frac{1}{\hbar}\int_{\gamma_{(\ominus,\uparrow)}}\Sigma[x_{cl},x_q]d\,\mathrm{Im}\,z} \tag{130}
$$
$$
=\exp\frac{2}{\hbar}\int_{-\hbar\lambda/2}^{0}d\tau\left\{\frac{m}{2}\left[\frac{dx_{cl\uparrow}(\tau)}{d\tau}\right]^2+V[x_{cl\uparrow}(\tau)]\right\}
$$
$$
\times\exp\frac{-2}{\hbar}\int_{-\hbar\lambda/2}^{\beta\hbar/2}d\tau\left\{\frac{m}{2}\left[\frac{dx_{cl\downarrow}(\tau)}{d\tau}\right]^2+V[x_{cl\downarrow}(\tau)]\right\}
$$
$$
\times\sqrt{\frac{1}{4\cosh(\hbar\omega\lambda/2)\cosh(\hbar\omega(\lambda+\beta)/2)}}\exp\left\{\frac{p_{cl}^2(t_f)\tilde{m}\omega}{2\hbar}\tanh(\hbar\omega\lambda/2)\right\}
$$
$$
\times\exp\left\{\frac{-p_{cl}^2(0)\tilde{m}\omega}{2\hbar}\tanh(\hbar\omega(\lambda+\beta)/2)\right\}.
$$

### G.4  semiclassical limit in the work functional

Above relation leads to the semiclassical definition of the work function up to the second order in $\hbar$. For a generic potential we write the fields in the vertical branch as $x_{cl\uparrow}(\tau) = x_{cl}(t_f) + \delta x_{cl\uparrow}(\tau)$ and keep only the second order in $\delta x_{cl\uparrow}(\tau)$

$$
\begin{aligned}
&\int \mathcal{D}x_{cl\uparrow} \exp \frac{2}{\hbar} \int_{-\hbar\lambda/2}^{0} d\tau \left\{ \frac{m}{2}\left[ \frac{dx_{cl\uparrow}(\tau)}{d\tau} \right]^2 + V[x_{cl\uparrow}(\tau)] \right\} \\
&= \int \mathcal{D}\delta x_{cl\uparrow} \exp \frac{2}{\hbar} \int_{-\hbar\lambda/2}^{0} d\tau \left\{ \frac{m}{2}\left[ \frac{d\delta x_{cl\uparrow}(\tau)}{d\tau} \right]^2 + V[x_{cl}(t_f)] + V'[x_{cl}(t_f)]\delta x_{cl\uparrow}(\tau) \right. \\
&\qquad\qquad\qquad \left. + \frac{1}{2}V''[x_{cl}(t_f)]\delta x_{cl\uparrow}(\tau)^2 \right\} \\
&= \exp \lambda \left\{ V[x_{cl}(t_f)] - \frac{V'[x_{cl}(t_f)]^2}{2V''[x_{cl}(t_f)]} \right\} \\
&\qquad \times \int D\delta x_{cl\uparrow} \exp \frac{1}{\hbar} \int_{-\hbar\lambda/2}^{0} d\tau \left\{ m\left[ \frac{\delta x_{cl\uparrow}(\tau)}{d\tau} \right]^2 + V''[x_{cl}(t_f)]\delta x_{cl\uparrow}(\tau)^2 \right\}. \quad (131)
\end{aligned}
$$

To obtain the last line we have completed the square in the last two terms of the second line and then redefined $\delta x_{cl\uparrow} + V'[x_{cl}]/V''[x_{cl}] \to \delta x_{cl\uparrow}$. Note that in this passage one should be careful with the integration domain of the path integral $\int D\delta x_{cl\uparrow}$. Before doing the change of variable the limit of integration was $\int_{\delta x_{cl\uparrow}(-\hbar\lambda/2)}^{\delta x_{cl\uparrow}(0)} D\delta x_{cl\uparrow}$. Using $\delta x_{cl\uparrow}(\tau) = x_{cl\uparrow}(\tau) - x_{cl}(t_f)$ we will have $\int D\delta x_{cl\uparrow} = \int_{0}^{x_{cl\uparrow}(0)-x_{cl}(t_f)} D\delta x_{cl\uparrow}$. Therefore the change of variable in the above equation will result in the change of domain into

$$
\int_{0}^{x_{cl\uparrow}(0)-x_{cl}(t_f)} D\delta x_{cl\uparrow} \to \int_{\frac{V'[x_{cl}(t_f)]}{V''[x_{cl}(t_f)]}}^{x_{cl\uparrow}(0)-x_{cl}(t_f)+\frac{V'[x_{cl}(t_f)]}{V''[x_{cl}(t_f)]}} D\delta x_{cl\uparrow}. \quad (132)
$$

With this in mind (131) can be written as

$$
\begin{aligned}
&\exp \lambda \left\{ V[x_{cl}(t_f)] - \frac{V'[x_{cl}(t_f)]^2}{2V''[x_{cl}(t_f)]} \right\} \\
&\qquad \times \int D\delta x_{cl\uparrow} \exp \frac{1}{\hbar} \int_{-\hbar\lambda/2}^{0} d\tau \left\{ m\left( \frac{d\delta x_{cl\uparrow}(\tau)}{d\tau} \right)^2 + V''[x_{cl}(t_f)]\delta x_{cl\uparrow}(\tau)^2 \right\} \\
&= \sqrt{\frac{1}{2\cosh\big(\hbar\omega(t_f)\lambda/2\big)}} \exp \lambda \left\{ V[x_{cl}(t_f)] - \frac{V'[x_{cl}(t_f)]^2}{2V''[x_{cl}(t_f)]} \right\} \\
&\qquad \times \exp \left\{ \frac{m\omega(t_f)}{\hbar} \left( \frac{V'[x_{cl}(t_f)]}{V''[x_{cl}(t_f)]} \right)^2 \tanh \hbar\omega(t_f)\lambda/2 \right\}.
\end{aligned} \quad (133)
$$

For the other vertical branch we will have

$$\int Dx_{cl\downarrow} \exp\frac{-2}{\hbar}\int_{-\hbar\lambda/2}^{\beta\hbar/2} d\tau \left\{\frac{m}{2}\left[\frac{dx_{cl\downarrow}(\tau)}{d\tau}\right]^2 + V[x_{cl\downarrow}(\tau)]\right\} \tag{134}$$

$$= \int D\delta x_{cl\downarrow}\exp\frac{-2}{\hbar}\int_{-\hbar\lambda/2}^{\beta\hbar/2} d\tau$$

$$\times\left\{\frac{m}{2}\left[\frac{\delta x_{cl\downarrow}(\tau)}{d\tau}\right]^2 + V[x_{cl}(0)] + V'[x_{cl}(0)]\delta x_{cl\downarrow}(\tau) + \frac{1}{2}V''[x_{cl}(0)]\delta x_{cl\downarrow}(\tau)^2\right\}$$

$$= e^{\left\{-(\lambda+\beta)\left[V[x_{cl}(0)]-\frac{[V'[x_{cl}(0)]^2}{2V''[x_{cl}(0)]}\right]\right\}}$$

$$\times\int D\delta x_{cl\downarrow}\exp\frac{-1}{\hbar}\int_{-\hbar\lambda/2}^{\beta\hbar/2} d\tau\left\{m\left[\frac{\delta x_{cl\downarrow}(\tau)}{d\tau}\right]^2 + V''[x_{cl}(0)]\delta x_{cl\downarrow}(\tau)^2\right\}$$

$$= \sqrt{\frac{1}{2\cosh(\hbar\omega(0)(\lambda+\beta)/2)}}\exp\left\{-(\lambda+\beta)\left[V[x_{cl}(0)]-\frac{V'[x_{cl}(0)]^2}{2V''[x_{cl}(0)]}\right]\right\}$$

$$\times\exp\left\{\frac{m\omega(0)}{\hbar}\left[\frac{V'[x_{cl}(0)]}{V''[x_{cl}(0)]}\right]^2\tanh(\hbar\omega(0)(\lambda+\beta)/2)\right\}.$$

Therefore (130) can be rewritten as

$$\int Dx_{cl}\int Dx_q Dp_q Dp_{cl}e^{\frac{1}{\hbar}\int_{\gamma_{(\ominus,\downarrow),(\ominus,M)}}\Sigma[x_{cl},x_q]d\operatorname{Im}z}e^{\frac{1}{\hbar}\int_{\gamma_{(\ominus,\uparrow)}}\Sigma[x_{cl},x_q]d\operatorname{Im}z} \tag{135}$$

$$= \frac{1}{2\cosh(\hbar\omega(t_f)\lambda/2)}\exp\left\{\frac{p_{cl}^2(t_f)\tilde{m}\omega(t_f)}{2\hbar}\tanh\hbar\omega(t_f)\lambda/2\right\}$$

$$\times\exp\lambda\left[V[x_{cl}(t_f)]-\frac{V'[x_{cl}(t_f)]^2}{2V''[x_{cl}(t_f)]}\right]$$

$$\times\exp\left\{\frac{m\omega(t_f)}{\hbar}\left[\frac{V'[x_{cl}(t_f)]}{V''[x_{cl}(t_f)]}\right]^2\tanh(\hbar\omega(t_f)\lambda/2)\right\}$$

$$\times\frac{1}{2\cosh(\hbar\omega(0)(\lambda+\beta)/2)}\exp\left\{\frac{-p_{cl}^2(0)\tilde{m}\omega(0)}{2\hbar}\tanh(\hbar\omega(0)(\lambda+\beta)/2)\right\}$$

$$\times\exp\left\{-(\lambda+\beta)V[x_{cl}(0)]-\frac{V'[x_{cl}(0)]^2}{2V''[x_{cl}(0)]}\right\}$$

$$\times\exp\left\{\frac{m\omega(0)}{\hbar}\left[\frac{V'[x_{cl}(0)]}{V''[x_{cl}(0)]}\right]^2\tanh(\hbar\omega(0)(\lambda+\beta)/2)\right\}.$$

The expansion of the above result is shown in (61) which indicates the semiclassical limit of work functional up to the second order in $\hbar$. This result is evidently satisfying the fluctuation relations.

## H  Work MGF for a harmonic oscillator

For the case of a time dependent harmonic oscillator the potential energy assumes the simple form as $V[x,t] = 1/2M\omega(t)x^2$ which for the vertical lines reduces to the intial $\omega(0) = \omega_0$ and final $\omega(t_f) = \omega_1$ value for the left and right vertical branches, respectively. For this potential, the two parameters $\omega(t_f)$ and $m(t_f)$ will be constant so the approximation

$V''[x_{cl\uparrow}(\tau)] \approx V''[x_{cl}(t_f)]$ is exact. Therefore using (130) and doing the path integral over $x_{cl}$ we will have

$$\int Dx_{cl} \int Dx_q Dp_q D_{p_{cl}} e^{\frac{1}{\hbar}\int_{\gamma_{(\Theta,\downarrow),(\Theta,M)}} \Sigma[x_{cl},x_q]d\,\mathrm{Im}\,z} e^{\frac{1}{\hbar}\int_{\gamma_{(\Theta,\uparrow)}} \Sigma[x_{cl},x_q]d\,\mathrm{Im}\,z} = \tag{136}$$

$$\times \sqrt{\frac{1}{2\cosh(\hbar\omega_1\lambda/2)}} \exp\left\{\frac{x_{cl}^2(t_f)m\omega_1}{\hbar\cosh(\hbar\omega_1\lambda/2)}\sinh(\hbar\omega_1\lambda/2)\right\}$$

$$\times \sqrt{\frac{1}{2\cosh(\hbar\omega_0(\lambda+\beta)/2)}} \exp\left\{\frac{-x_{cl}^2(0)m\omega_0}{\hbar\cosh(\hbar\omega_0(\lambda+\beta)/2)}\sinh(\hbar\omega_0(\lambda+\beta)/2)\right\}$$

$$\times \sqrt{\frac{1}{2\cosh\hbar\omega_1\lambda/2}} \exp\left\{\frac{p_{cl}^2(t_f)}{m\omega_1\hbar\cosh(\hbar\omega_1\lambda/2)}\sinh(\hbar\omega_1\lambda/2)\right\}$$

$$\times \sqrt{\frac{1}{2\cosh(\hbar\omega_0(\lambda+\beta)/2)}} \exp\left\{\frac{-p_{cl}^2(0)}{m\omega_0\hbar\cosh(\hbar\omega_0(\lambda+\beta)/2)}\sinh(\hbar\omega_0(\lambda+\beta)/2)\right\}.$$

To go further we need to integrate over $p(t_f)$, $x(t_f)$, $p_{cl}(0)$ and $x_{cl}(0)$, However, the relation between them is given by the path integral over the horizontal line that results in the equation of motion for a time dependent harmonic oscillator as $x(t_f) = A(t_f)x(0) + B(t_f)p(0)/M$ and $p(t_f) = M\dot{A}(t_f)x(0) + \dot{B}(t_f)p(0)$. Therefore we can write the above relation as

$$M_W(\lambda,t) = \frac{\sinh(\beta\hbar/2)}{2\cosh(\hbar\omega_1\lambda/2)\cosh(\hbar\omega_0(\lambda+\beta)/2)} \exp\left\{\frac{-1}{2}\begin{pmatrix} x(0) & p(0) \end{pmatrix}\begin{pmatrix} C_{11} & C_{12} \\ C_{21} & C_{22} \end{pmatrix}\begin{pmatrix} x(0) \\ p(0) \end{pmatrix}\right\}, \tag{137}$$

with

$$C_{11} = \frac{2m\omega_0}{\hbar}\tanh(\hbar\omega_0(\lambda+\beta)/2) - \frac{2m}{\hbar\omega_1}\left[A^2\omega_1^2 + \dot{A}^2\right]\tanh(\hbar\omega_1\lambda/2),$$

$$C_{22} = \frac{2}{m\omega_0\hbar}\tanh(\hbar\omega_0(\lambda+\beta)/2) - \frac{2}{\hbar m\omega_1}\left[\omega_1^2 B^2 + \dot{B}^2\right]\tanh(\hbar\omega_1\lambda/2), \tag{138}$$

$$C_{12} = C_{21} = -\left[\frac{2\omega_1}{\hbar}AB + \frac{2}{\omega_1\hbar}\dot{A}\dot{B}\right]\tanh(\hbar\omega_1\lambda/2),$$

where the term $\sinh\beta\hbar/2$ comes form the normalization of the generating function. Therefore the integration over $p_0$ and $x_0$ will result in $2\pi/\sqrt{\det(\mathbf{C})}$ which can be written as

$$M_W(\lambda,t_f) = \frac{\sinh(\beta\hbar/2)}{\cosh(\hbar\omega_1\lambda/2)\cosh(\hbar\omega_0(\lambda+\beta)/2)} \tag{139}$$

$$\times \frac{1}{\sqrt{\tanh^2(\hbar\omega_0(\lambda+\beta)/2) - 2Q^*\tanh(\hbar\omega_1\lambda/2)\tanh(\hbar\omega_0(\lambda+\beta)/2) + \left[\dot{A}B - \dot{B}A\right]^2\tanh^2(\hbar\omega_1\lambda/2)}}.$$

We notice that the last term under the square root is the Wronskian and for our given boundary conditions is 1. Also for $Q^*$ we have

$$Q^* = \frac{1}{\omega_0\omega_1}\left\{\omega_0^2\left[\omega_1^2 B^2 + \dot{B}^2\right] + \left[\omega_1^2 A^2 + \dot{A}^2\right]\right\}. \tag{140}$$

Thus we write (139) as

$$M_W(\lambda,t_f) = \frac{\sqrt{2}\sinh(\beta\hbar/2)}{\sqrt{\cosh(\hbar\omega_0(\lambda+\beta))\cosh(\hbar\omega_1\lambda) - Q^*\sinh(\hbar\omega_0(\lambda+\beta))\sinh(\hbar\omega_1\lambda) - 1}}. \tag{141}$$

This allows us to recover the results of ref [72].

# I   Classical limit in detailed balance conditions

In Sec. 5.4 we introduced a notion of detailed balance at the level of the quantum trajectories. The strategy of Sec. 5.4 is to divide the path integration in Eq. (48) in three separate propagators, that read

$$\mathcal{E}_{\lambda,t_f}[y_q^f, y_{cl}^f, \alpha(t_f)] = \frac{1}{\lambda} \log \int_{B(y^f)} \mathcal{D}' x_{cl/q} e^{\frac{1}{\hbar} \int_{\gamma_{(\ominus,\uparrow)}} \Sigma[x_{cl}(z), x_q(z)] d \operatorname{Im} z}, \tag{142}$$

$$\mathcal{E}_{-\lambda-\beta,0}[y_q^i, y_{cl}^i, \alpha(0)] = \frac{1}{-\lambda-\beta} \log \int_{B'(y^i)} \mathcal{D}' x_{cl/q} e^{\frac{1}{\hbar} \int_{\gamma_{(\ominus,\downarrow),(\ominus,M)}} \Sigma[x_{cl}(z), x_q(z)] d \operatorname{Im} z}, \tag{143}$$

$$\mathcal{U}[y_q^f, y_{cl}^f, t_f; y_q^i, y_{cl}^i, 0] = \int_{B''(y^i, y^f)} \mathcal{D}' x_{cl/q} e^{\frac{i}{\hbar} \int_{\gamma_{(\ominus,-)}} \mathcal{M}[x_{cl}(z), x_q(z)] d \operatorname{Re} z}, \tag{144}$$

where $B(y^i), B'(y^f), B''(y^i, y^f)$ are shortcuts to denote the boundary conditions of the path integrals. For Eq. (142) the boundary conditions are $x_{q\uparrow}(-\hbar\lambda/2) = y_q^f, x_{q\uparrow}(0) = 0, x_{cl\uparrow}(-\hbar\lambda/2) = y_{cl}^f$, for Eq. (143) we have[1] $x_{q\downarrow}(\hbar\beta/2) = 0, x_{q\downarrow}(-\hbar\lambda/2) = y_q^i, x_{cl\downarrow}(-\hbar\lambda/2) = y_{cl}^i$ while for Eq. (144) we have $x_q(0), x_{cl}(0) = y_q^i, y_{cl}^i, x_q(t_f), x_{cl}(t_f) = y_q^f, y_{cl}^f$. The definition of the time-reversed trajectory is arbitrary, but similarly to what we did in Sec. 4.3, for every trajectory defined in the Keldysh contour $\gamma_K$, we can consider the trajectory that attains the same values but on a reversed contour in which the forward and backward branches are exchanged. This trajectory starts in the final points $(y_{cl}^f, -y_q^f)$ and ends in the initial points $(y_{cl}^i, -y_q^i)$, where the sign of the quantum components has been changed due to the inversion of the forward and backward branches. We have $\mathcal{U}(y_q^f, y_{cl}^f, y_q^i, y_{cl}^i) = \mathcal{U}^{\text{rev}}(-y_q^i, y_{cl}^i, -y_q^f, y_{cl}^f)$ from which Eq. (66) follows.

Let us now focus on the classical expansion of the quantity $\mathcal{E}_{-\beta}$, that is the integral in the branch $\gamma_{(\ominus,M)}$. The action in this case is given by $\Sigma$, that can be expanded for small $x_{q\downarrow}$[2] as

$$\Sigma = m \left( \dot{x}_{cl\downarrow}^2 + \dot{x}_{q\downarrow}^2 \right) + 2V(\alpha, x_{cl\downarrow}) + V''(\alpha, x_{cl\downarrow}) x_{q\downarrow}^2 + O(\hbar^2). \tag{145}$$

Note that respect to Eq. (49) there is a change of sign in the kinetic term, due to the use of $x_{cl\downarrow}(\tau)$ instead of $\gamma_{cl}(z)$, since we have $\frac{d}{dz} = -i\frac{d}{d\tau}$ in the vertical branches. Since the branch $\gamma_{(\ominus,\downarrow)}$ is short for $\hbar \to 0$, the value of $x_{cl\downarrow}$ will not variate too much from the value attained at the boundary with $\gamma_{(\ominus,-)}$ (that is $y_{cl}^i$) so we can define $x_{cl\downarrow}(s) = y_{cl}^i + \delta(s)$ and obtain

$$\Sigma = m \left( \dot{\delta}^2(s) + \dot{x}_{q\downarrow}^2(s) \right) + 2V(\alpha(0), y_{cl}^i) + 2V'(\alpha(0), y_{cl}^i)\delta(s)$$
$$+ V''(\alpha(0), y_{cl}^i)[x_{q\downarrow}^2(s) + \delta^2(s)] + O(\hbar^2). \tag{146}$$

We are mainly interested in the case in which the boundaries in the path integration in Eq. (143) with $\lambda = 0$ are given by $x_{q\downarrow}(\hbar\beta/2) = x_{q\downarrow}(0) = 0$, a case in which the contribution of the integral over the quantum variables is negligible. The path integration in $x_{cl\downarrow}$ reduces to a path integration in $\delta(s)$ with boundary conditions $\delta(0) = 0$, while the initial value of $\delta$ is free, we call it $\bar{\delta}$:

$$\int \mathcal{D}' \delta e^{\frac{1}{\hbar} \int_{\frac{\hbar\beta}{2}}^{0} d\tau \left( m\dot{\delta}^2(s) + 2V(\alpha(0), y_{cl}^i) + 2V'(\alpha(0), y_{cl}^i)\delta(s) + V''(\alpha(0), y_{cl}^i)\delta^2(s) \right)}$$

$$= \int \mathcal{D}' \delta e^{\frac{1}{\hbar} \int_{\frac{\hbar\beta}{2}}^{0} d\tau \left[ m\dot{\delta}^2(s) + m\Omega^2 \left( \delta + \Delta \right)^2 - m\Omega^2\Delta^2 \right]}, \tag{147}$$

---

[1] Here we use the notation $x_{q/cl\downarrow}(\tau)$ for the fields on the branch $\gamma_{(\ominus,\downarrow),(\ominus,M)}$ (see Eq (123)).
[2] See footnote 1.

where we introduced $\Omega = \sqrt{\frac{V''(\alpha(0), y_{cl}^i)}{m}}$ and $\Delta = \frac{V'(\alpha(0), y_{cl}^i)}{V''(\alpha(0), y_{cl}^i)}$. After doing the change of variable $\delta \to \delta + \Delta$ and solving the path integral (remember that changing the variable also changes the boundaries of the path integration) we obtain as a result

$$
e^{-\beta[V(\alpha(0), y_{cl}^i) - \frac{1}{2} m \Omega^2 \Delta^2]} \left( \frac{m\Omega}{\pi\hbar \sinh\left(\frac{\beta\hbar\Omega}{2}\right)} \right)^{\frac{1}{2}} \exp\left\{ -\frac{m\Omega}{\hbar} \frac{\cosh\left(\frac{\hbar\beta\Omega}{2}\right)[(\bar\delta + \Delta)^2 + \Delta^2] - 2\Delta(\bar\delta + \Delta)}{\sinh\left(\frac{\hbar\beta\Omega}{2}\right)} \right\}
$$

$$
\approx e^{-\beta[V(\alpha(0), y_{cl}^i) - \frac{1}{2} m \Omega^2 \Delta^2]} \left( \frac{2m}{\pi\hbar^2\beta} \right)^{\frac{1}{2}} \exp\left\{ -\frac{2m}{\hbar^2\beta}\bar\delta^2 - \frac{2m}{\hbar^2\beta} \frac{\hbar^2\beta^2\Omega^2}{8}[2\Delta^2 + 2\bar\delta\Delta + \bar\delta^2] \right\}, \quad (148)
$$

where the two contributions in the last exponential come from the zeroth and second order expansions of the hyperbolic cosine in respect to $\hbar$. Now note that the terms proportional to $\Delta^2$ cancels out so we are left with the integral over $\bar\delta$

$$
\int_{-\infty}^{\infty} d\bar\delta\, e^{-\beta V(\alpha(0), y_{cl}^i)} \left( \frac{2m}{\pi\hbar^2\beta} \right)^{\frac{1}{2}} \exp\left\{ -\frac{2m}{\hbar^2\beta}\bar\delta^2 - \frac{m\beta^2\Omega^2}{4}[2\bar\delta\Delta + \bar\delta^2] \right\}
$$

$$
\approx \int_{-\infty}^{\infty} d\bar\delta\, e^{-\beta V(\alpha(0), y_{cl}^i)} \left( \frac{2m}{\pi\hbar^2\beta} \right)^{\frac{1}{2}} \exp\left\{ -\frac{2m}{\hbar^2\beta}\bar\delta^2 \right\} = e^{-\beta V(\alpha(0), y_{cl}^i)}, \quad (149)
$$

where the second term in the integrand in the first line can be neglected since it is of the next order in $\hbar$. This proves the result (67). To prove the last result it is sufficient to use the properties of the Wigner function for small $\hbar$, so we can focus on Eq. (65).

$$
\iint dy_q^i dy_q^f K(y_q^f, y_{cl}^f, y_q^i, y_{cl}^i) \tag{150}
$$

$$
= \iint dy_q^i dy_q^f \iint dp^i dp^f \iint d\eta^i d\eta^f K(\eta^f, y_{cl}^f, \eta^i, y_{cl}^i) e^{ip^i(y_q^i - \eta^i)} e^{ip^f(y_q^f - \eta^f)}.
$$

Let us focus on the dependence by $\eta^i, y_q^i, p^i$, after replacing the definition of $K$ in the equation above we have a contribution of the form

$$
Z(0)^{-1} \iint d\eta^i dy_q^i e^{ip^i(y_q^i - \eta^i)} \mathcal{U}[\eta^f, y_{cl}^f, t_f; \eta^i, y_{cl}^i, 0] e^{-\beta \mathcal{E}_{-\beta}[\eta^i, y_{cl}^i, \alpha(0)]}. \tag{151}
$$

Remembering the definition of $\mathcal{E}_{-\beta}$ we can also write

$$
e^{-i\eta^i p^i} Z(0)^{-1} e^{-\beta \mathcal{E}_{-\beta}[\eta^i, y_{cl}^i, \alpha(0)]} = e^{-i\eta^i p^i} \left\langle y_{cl}^i - \eta^i \middle| \rho_0 \middle| y_{cl}^i + \eta^i \right\rangle. \tag{152}
$$

After performing the integral in $\eta^i$, the quantity above becomes (up to irrelevant prefactors) the Wigner function associated to the initial state, that we denoted with $W_\beta$. In a similar way, we can show that the integral over $y_q^i$ and the integrals over $y_q^f$ transform the propagator $\mathcal{U}$ in the quantum phase space propagator.

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
