# Peer review of "A convenient Keldysh contour for thermodynamically consistent perturbative and semiclassical expansions"

_SciPost Physics, doi:SciPost Phys. 15, 209 (2023)_

## Round 1 · Referee Report · Anonymous (Referee 1) · 2023-7-11

Strengths

This work establishes a connection between the fluctuation theorem in the quantum and semiclassical regimes by utilizing the Keldysh non-equilibrium diagrammatic expansion and the Keldysh path-integral approach based on the geometric symmetry of novel modified Keldysh contours.

Weaknesses

The current version appears to have several ambiguities. It would be preferable to clarify the connections to previous related works.

Report

In the paper, the authors discuss the thermodynamic consistency of the Keldysh formalism in the perturbative and path-integral approaches. The novel ingredients of their work seem to be the modified contour Fig. 1c and the symmetrized contour Fig. 4c. The modified contour allows for the interpretation of the fluctuation theorem (FT) for the generating function based on the geometrical symmetry, as illustrated in Fig. 2. By performing the perturbative expansion, the authors also highlight that the work statistics can be understood as the summation of Poisson processes, as shown in Eq. (33). If I understand correctly, the symmetrized contour enables the expression of 'stochastic' work in a trajectory-independent form, as demonstrated in Eq. (52), and facilitates the derivation of the detailed balance relation at the level of quantum trajectories, as exemplified by Eq. (66).

The paper provides a sufficiently concise explanation of the theoretical framework. Technical details, along with several explicit examples, are provided in the appendices. The topic of the thermodynamically consistent Keldysh non-equilibrium field theory is important, and the paper aims to offer an interesting interpretation of the FT based on the geometrical symmetry of properly modified Keldysh contours. Therefore, I believe the paper is suitable for publication. However, I have some questions and comments that I hope will contribute to further improving the paper.

Requested changes

1) Is the contour introduced in the present paper in Fig. 1c equivalent to that introduced previously, for example, in Section 8 of H. Umezawa's book, "Advanced Field Theory: Micro, Macro, and Thermal Physics" (AIP, New York, 1993)?
2) The time-dependent Hamiltonian defined at the 'complex time,' e.g., the sinusoidal driving with frequency \Omega, H_1(z) \propto \sin(\Omega z), does not make sense. Therefore, it is better to explain the definition of H_1(z) more carefully.
3) In Sec. 4.2, the authors found that the work statistics become the sum of independent rescaled Poisson processes, Eq. (33). I believe this holds for the lowest-order diagrams. However, when one performs an infinite summation, it is known that the statistics change for the charge full-counting statistics (See, e.g., Eq. (9) of Ref. [26]). I would expect a similar behavior for the work statistics.
4) In Appendix C, the authors calculated the d=1,2,3,4 diagrams in Fig. 3b. It is unclear why the contributions of d=1,2 diagrams vanish. Additionally, the origin of (..)|_{\lambda=0} terms in the d=3,4 diagrams in Eq. (86) is not clear either. I speculate that these terms may originate from the d=1,2 diagrams.
5) The authors should also provide an explicit derivation of the FT at the level of the cumulant generating function using Eqs. (33) and (37). Additionally, it would be helpful to clarify the connection between E_d and E^d introduced above Eq. (32).
6) The authors discovered that the FT holds at every order in the expansion of the generating function. This finding has been utilized in previous literature on the FT based on Keldysh diagrams [e.g., Saito and Utsumi, "Symmetry in Full Counting Statistics, Fluctuation Theorem, and Relations among Nonlinear Transport Coefficients in the Presence of a Magnetic Field," arXiv:0709.4128; Phys. Rev. B 78, 115429 (2008); Utsumi and Saito, "Fluctuation Theorem in a Quantum-Dot Aharonov-Bohm Interferometer," Phys. Rev. B 79, 235311 (2009); Utsumi, Entin-Wohlman, Ueda, Aharony, "Full-counting statistics for molecular junctions: Fluctuation theorem and singularities," Phys. Rev. B 87, 115407 (2013)]. It would be valuable for the authors to comment on the novel developments achieved compared to the previous works.
7) The authors state that the 'stochastic' work becomes trajectory independent in the classical limit (\hbar \to 0) in Eq. (60). However, this contradicts Eq. (56) and the fact that the stochastic work in the stochastic thermodynamics is typically trajectory dependent. A more thorough explanation is necessary to resolve this apparent contradiction.
8) Besides the references by Funo and Quan [22, 23], theFT based on the path integral in the semiclassical limit and the FT in the classical limit based on the Martin-Siggia-Rose-Janssen-deDominicis action have been developed [e.g., Mallick, Moshe, Orland, "A field-theoretic approach to non-equilibrium work identities," J. Phys. A: Math. Theor. 44, 095002 (2011); Utsumi, Golubev, Marthaler, Schön, Kobayashi, "Work fluctuation theorem for a classical circuit coupled to a quantum conductor," Phys. Rev. B 86, 075420 (2012); Aron, Barci, Cugliandolo, González Arenas, Lozano, "Dynamical symmetries of Markov processes with multiplicative white noise," J. Stat. Mech. 053207 (2016)]. It would be enlightening to compare the present work with these previous theories. 
9) In the third line of Eq. (16), there might be an error in the ordering of U^\dagger and U.
10) Before Eq. (18), the authors wrote, "With this in mind, ... of the contour itself as shown in Fig. 3." Should it be Fig. 2 instead of Fig. 3?
11) The definition of the step function in Eq. (27) should be provided.
12) Below Eq. (37), the authors wrote C_W(\lambda,t_f) = C_W^{\rm rev}(\lambda,t_f), which appears to be inconsistent with Eq. (17).
13) In the figure captions and the main text, the authors sometimes use small letters to indicate specific figures, such as Fig. 1a. It would be preferable to use capital letters, as they are used in the figures. Additionally, before Eq. (79), the authors wrote "... the results of App. (B)..." while below Eq. (84), the authors wrote "... in detail in App. D." It would be preferable to maintain consistent notation throughout the paper.
14) In Fig. 2D, t_f might be t_f + I \hbar \lambda.
15) Below Eq. (62), it appears that x_{q \uparrow}(-\hbar \lambda/2) should be x_{q \uparrow} (-i \hbar \lambda/2) based on the definition in Eq. (47).
16) In Eq. (68), the authors introduced the term W_{cl}. Is it the same as W_0(t_f) defined before Eq. (61)?
17) Before Eq. (121), the authors wrote "... and replacing the expressions (120) and (121) ...". This statement may be incorrect.
18) Since there are a lot of equations and I have not checked all of them, it would be advisable for the authors to recheck all derivations.

---

## Round 1 · Referee Report · Anonymous (Referee 2) · 2023-7-20

Strengths

This is a very nice work, and I strongly recommend it for publication.

Weaknesses

No significant weaknesses.

Report

This is a very nice work, and I strongly recommend it for publication. The search for a proof of thermodynamic consistency within the formalism of Keldysh Green's functions has been a goal of quantum thermodynamics for many years. This work appears to resolve much of the problem. This work is sufficiently deep that it will take me more time to *fully* understand it than is reasonable for a refereeing process. However, I make some suggestions for changes below, based on my current understanding.

Requested changes

I have three suggestions for changes that would make the work easier to follow (based on my current understanding). I suggest that the authors implement those suggestions that make sense.

(1) If I understand correctly this work proves thermodynamic consistency for a work fluctuation theorem in a situation with a single temperature. That would mean that the proof applies to the thermodynamics of situations in which one is doing work on the system (on average), and that work is dissipated as heat (on average). But it does not apply to situations in which a temperature difference is used to create work. If so, I think it would be worth saying this to the reader in the introduction. If I am mistaken about this, then the authors should definitely add a discussion of this point in the manuscript. In any case, this work is a big step forward.

(2) I found it very hard to follow the crucial text between Eq. (35) and (36). This is the text that explains how to identify the diagram that is the "rev" partner of another diagram. I feel a figure with a simple example (or two) would help readers understand quickly. For example, what is the "rev" partner of the diagram in Fig 3 or the dumbbell in Fig 6?
Indeed, it would be great if there is a simple graphical rule to identify the "rev" partner of a given diagram. This would be analogous to Whitney's rule (in the context of a different type of Keldysh method in the manuscript's Ref [31]) in which one simply rotates a diagram by 180 degrees to find its time-reversed partner. However, I would understand if there is no such simple rule here.

(3) VERY MINOR COMMENT: I found the acronyms GF, FT, MGF and CGF to be distracting, and hard to remember while concentrating on following the mathematics. This was particularly so because the G and F are different in GF, FT and MGF. As there is no page limit, I think readers will be pleased if all these acronyms are replaced by the full words throughout the manuscript.

---

## Round 2 · Referee Report · Anonymous (Referee 2) · 2023-10-13

Report

The authors have convincingly responded to all my comments/questions. I now recommend this manuscript for publication without hesitation.

---

## Round 2 · Referee Report · Anonymous (Referee 1) · 2023-10-22

Report

The authors have satisfactorily addressed my previous questions and comments in the rebuttal letter. They have also extended the explanations in the main text accordingly. In addition, they have provided detailed explanations in the rebuttal letter regarding the connection to previous works, which would be valuable information for potential readers. This connection is not as prominently reflected in the main text, which somewhat weakens the paper. Despite this, I believe the revised paper contains valuable materials for publication.

---

## Round 2 · Author Response

REPLY TO REFEREE 1

We would like to thank Referee 1 for their positive assessment of our manuscript and for providing useful comments and suggestions. We are pleased to read that Referee 1 strongly recommends the paper for publication and states that the paper does not present significant weaknesses. In the following we would like to provide answers to the three points raised by Referee 1:

(1) We thank the referee for pointing out this fact. The referee is right: in our approach we only described the work generation in a single closed system with fixed initial temperature. This encompasses all the cases in which a closed system is subjected to an external driving that is capable of changing the state of the system and altering the energy stored into it. The situation is more complicated when we deal with a system coupled with many baths at different temperatures. In this case there is no unique way to introduce the modified contour in Fig. 1C because there are many temperatures in the problem, and it is not straightforward to assign a length to the vertical branch corresponding to the initial preparation. In addition, in the typical open system scenario one is interested not only in probing the energy variation in the system but also in all the reservoirs (these energy variations correspond to the heat flows). Hence, in this scenario it is usually not sufficient to use a single counting field \lambda, since we have to use a different counting field for each measured observable (see for instance reference [35] in the revised version of the manuscript). As the referee correctly points out, in this scenario a possible source of work is not only given by the driving, but also from the difference of the heat flows between the baths. We believe that analyzing how to generalize our framework to a case with many baths would be an interesting follow up project, so we added a comment about this in the conclusions of the paper.

(2) We thank the referee for this useful comment. The discussion of the time-reversed diagrams was indeed unclear, so we decided to revise that part. Using the fact that in the contour of the time-reversed generating function the forward and backward branches are inverted, we can introduce the time-reversed diagram as a diagram with the same vertex coordinates as the original, but placed on the time-reversed contour \gamma_{rev} in Fig. 2 D. To connect this with the results of [31], we note that since the position of \gamma_- and \gamma_+ are inverted in the contour \gamma_{rev}, to implement our mapping we have to “flip” the vertex variables, since to stay in \gamma_- (or \gamma_+) they have to jump between the lower and upper branch (since in \gamma the forward branch is the lower branch, while in \gamma_{rev} is the upper branch). This is very similar to the result of [31] in which the mapping consists of a rotation of 180 degrees in the plane of the contour. In both mappings the vertex arguments jump from the upper to the lower branch. Since for computing the contribution of a given “charged” diagram we have to integrate the variables over the branch on which they are defined, we believe that (in our case) the exact mapping in itself is not relevant but only the fact that the mapping provides a one-to-one correspondence between \gamma_- in \gamma and \gamma_- in \gamma_{rev}, and the same for \gamma_+. For this reason, we believe that using the mapping of [31] should work perfectly also in our case.

(3) We have carefully reevaluated the use of acronyms in our manuscript. We decided to remove the acronym “CGF” and use “cumulant generating function” in the text. On the other hand, we prefer to continue to use the acronyms GF and MGF because they are standard acronyms and extensively used in the manuscript.

REPLY TO REFEREE 2

We would like to thank Referee 2 for stating that the paper is suitable for publication and for providing very useful comments and suggestions. The expert advise of the Referee has been of great help for us in improving the clarity of the presentation, the completeness of the bibliography, and the overall quality of the manuscript. In the following we try to answer in detail to the points raised by the Referee:

1) We thank the referee for pointing out this reference, it contains a very concise explanation of the relation between path-ordering and modified contours. The contour in chapter 8 is equivalent to the one in fig. 1c of the manuscript (and thus, equivalent to the ones in the cited references [22], [24] of the revised manuscript) after defining \lambda = (\alpha-1) \beta and including \hbar to uniform the physical dimensions. We added a reference to the book of H. Umezawa in the revised version of the manuscript.

2) H_1(z) is always assumed to be in the Schrodinger picture and follows the definition of the contour Hamiltonians (11), (13). As such, it is always constant on the vertical branches and it can be time-dependent only on the horizontal branches of the contour, since these branches define the domain of the physical time coordinates of the process. For a sinusoidal driving we would have H_1(z) \propto \sin(\Omega z) for z \in \gamma_{\pm}, i.e., on the horizontal branches, while on the vertical ones we would have the constant values corresponding to the initial and final perturbations H_1(0) \propto \sin(\Omega 0) = 0, H_1(t_f) \propto \sin(\Omega t_f). To stress this point, we added a reference to Eq. (13) after introducing H_1(z).

3) The referee is correct about this point. The work statistics will not be given anymore by a sum of rescaled Poisson processes in the limit in which an infinite number of diagrams is summed. The Poissonian statistics in the perturbative regime reflects the fact that perturbative interactions correspond to rare events. It remains true to higher orders in perturbation theory, but an increasingly long series of Poisson processes can of course give rise effectively to different statistics. This observation opens the way to a lot of interesting further developments of our work, since many of the results of many-body perturbation theory are obtained through a summation of specific subfamilies of diagrams. We added a comment in Sec. 4.2 to share this interesting insight with the readers.

4) We apologize for the confusion in the notation. The diagrams d=1,2 do not contribute because they have E_d=0, so the term in the parenthesis in Eq. (33) vanishes. This does not happen for the diagrams d=3,4, for which this term is given by the difference of two different quantities, one with \lambda \neq 0, and one with \lambda=0, as in Eq. (86). We added a comment above Eq. (33) to clarify this point and wrote a more thorough explanation before Eq. (86).

5) We added a sketch of the derivation of the FT at the level of diagrams at the end of Sec. 4.3. The superscript d was a typo that has now been corrected.

6) We thank the referee for pointing out these relevant references, they have all been cited in the new version of the manuscript. In the three references the statistics of the energy currents is computed in detail for a single quantum dot locally coupled to an Einstein phonon, for a multi-terminal device in presence of a magnetic field and for a two-terminal Aharonov-Bohm interferometer. The authors discuss the thermodynamic consistency of their approach in a detailed way (for instance, in “Full-counting statistics for molecular junctions: the fluctuation theorem and singularities” they introduce an Hartree approximation that preserve the validity of the fluctuation theorem). There are some connections between the first part of our manuscript and the three references mentioned above, since these references deal with a perturbative expansion of the CGF. However, we would like to point out that the derivation of FT contained in our manuscript is more general than the ones contained in the papers mentioned above, since it holds for any value of the final time and not only in the long-time limit. In addition, the proof of FT presented in the manuscript holds for the case in which the Hamiltonian has an arbitrary time dependence, with the final Hamiltonian being also possibly different from the initial one. In this last case a new family of GF components play a role as explained in appendix D. Finally, in the three papers mentioned by the referee the FT is proved without using the modified contour in our fig. 1C but instead relying on different techniques. For instance, in “Full-counting statistics for molecular junctions: the fluctuation theorem and singularities” the starting point is the counting field dependent Hamiltonian (13). Derivations of the FT using a counting-field dependent Hamiltonian can be done, for instance, in the case of quantum master equations (see reference [37] of the new version of the manuscript), but can be very challenging in other situations, like the path integral scenario used in the second part of the paper. On the contrary, our proof using the modified contour is completely general and can be applied regardless of the details of the Hamiltonian and the final time. In our opinion, it represents also an elegant way to show that the FT is a symmetry that is inherently present in the structure of non-equilibrium physics, since it is encoded in the contour.

7) We apologize for this part being unclear. What we mean by “trajectory independent” is that its dependence on any given trajectory reduces to a dependence on the endpoints and the initial points of the trajectory. This property crucially relies on the system being closed, while typically in stochastic thermodynamics the work cannot be reduced to a function of the endpoints of the system trajectory due to the presence of environment degrees of freedom. The contradiction between the classical limit Eq. (60) and the trajectory-dependent work (56) can be resolved by noting that in the classical limit Eq. (56) reduces to an integral of the instantaneous power, which is trajectory-dependent but can be reduced to Eq. (60) after performing the integral over time explicitly. We added a discussion about this point in the revised version of the manuscript.

8) We thank the Referee for pointing out these references. In "Dynamical symmetries of Markov processes with multiplicative white noise" and "A field-theoretic approach to non-equilibrium work identities" a theoretical derivation of the FT is presented. In these papers, the integral FT is derived as a symmetry of the average of the exponential of a quantity Delta S (that in the first of the two papers is connected to the MSRJD action, see Eq. 2.22). To connect this quantity to the physical entropy production, further assumptions are needed, as later explained in the two references (initial Gibbs preparation, detailed balance conditions). Starting from this observation, we point out that in the literature there are two complementary approaches to the derivation of the FT: in the first, a physical quantity of interest is identified (work, entropy production) and the symmetries of the forward and time reversed CGF of this quantity are discussed; in the second, a functional of the forward and time-reversed trajectories is introduced in order to satisfy a mathematical symmetry, that reduces to the FT for physical observables only after taking some further assumptions. In our manuscript, we follow the first path while the second one is chosen in the two references discussed above. In addition, we point out that these references prove the FT in a particular case (Langevin equation with delta-correlated gaussian noise) of classical stochastic thermodynamics, while our proof applies to an arbitrary closed quantum system. The third reference “Work fluctuation theorem for a classical circuit coupled to a quantum conductor” contains a derivation of FT in the case of a quantum conductor, driven by an external voltage source, and employs quantum path integral techniques and Keldysh rotation as a tool to perform a classical expansion of the CGF. We now cite this paper in the introduction.

9) Eq. (16) is correct. With respect to the second line, the ordering of U^{dag} and U is inverted, as correctly pointed out by the referee, but also the ordering of the exponential \lambda dependent terms is inverted, restoring the validity of the equality due to the cyclic property of the trace.

10) We corrected this typo in the new version of the manuscript.

11) We thank the Referee for noting that the definition of the contour step function was absent in the main text. We included a sentence to define \Theta_{\gamma} in Sec. 4.1.

12) We thank the Referee for noting this typo. In the second member of the equality it should be C_W^{rm rev}(-\lambda-\beta,t_f). The typo has now been corrected.

13) In the new version of the manuscript the notation has been made more uniform by removing the parenthesis from the references to appendices and using capital letters for referencing different panels in the figures.

14) We agree with the referee that there should be t_f + I \hbar \lambda in Fig. 2D. We have corrected this.

15) Below Eq. (62) the notation x_{q \uparrow}(-\hbar \lambda/2) is correct, but we understand that this notation could generate some confusion. For the sake of clarity, we added a comment on the notation in sec. 5.4.

16) The two terms are the same and the discrepancy has been corrected in the new version of the manuscript.

17) We thank the Referee for pointing out this. In the first version there was a misplaced reference that has now been corrected.

18) Following this suggestion, we carefully checked all the calculations. We would like to thank the referee again for the careful reading of our manuscript and for the helpful questions and comments.

---

## Round 2 · List of Changes

- The references [25], [32], [33] have been added in the introduction;
- The reference [39] has been added in the introduction, when discussing fluctuation theorems;
- The last paragraphs of the introduction have been modified;
- The references [49], [50, [51] have been added at the end of Sec. 3, together with a brief comment on modified contour techniques;
- The reference to the figure above Eq. (18) has been corrected;
- We added a clarification on the dependence by time of H_1(z) at the beginning of Sec. 4;
- We included the definition of Theta function in the contour below Eq. (27);
- We included a comment on the connection between the resummation of diagrams and the Poisson statistics at the end of Sec. 4.2;
- A derivation of the FT at the level of the cumulant generating function starting from Eq. (34) has been added to the text at the end of Sec. 4.3;
- We included a comment above Eq. (33) to make its meaning clearer;
- We corrected the typo below Eq. (37);
- We added a comment above Eq. (47) to say that our contour after symmetrization is similar to the one of [49];
- We added a clarification below Eq. (60) about the classical limit of the fluctuating work;
- We uniformed the notation for referencing to figures and to appendices;
- A new version of Fig. 2 has been used;
- A new version of Fig. 5 has been used;
- We added a more detailed explanation about the definition of time-reversed process above Eq. (66);
- In Eq. (68), we replaced W_{cl} with W_0;
- We extended the discussion before Eq. (86) in the appendix, for the sake of clarity;
- We corrected the references before Eq. (121) (Eq. (122) in the new version of the manuscript);
- We included in the conclusion a comment about possible further developments of our work in the open system framework;
- A comment to clarify the notation x_{q/cl\uparrow} has been added under Eq. (62).

---

## Editorial Decision

published